TOPICAL REVIEW

# The causative role of amyloidosis in the cardiac complications of Alzheimer's disease: a comprehensive systematic review

Samuel Parker[1] 🆔, Andrew F. James[2] 🆔 and Svetlana Mastitskaya[1] 🆔

[1]*Department of Translational Health Sciences, Bristol Medical School, University of Bristol, Bristol, UK*
[2]*School of Physiology, Pharmacology & Neuroscience, University of Bristol, Bristol, UK*

Handling Editors: Bjorn Knollmann & T Alexander Quinn

The peer review history is available in the Supporting Information section of this article (https://doi.org/10.1113/JP286599#support-information-section).

**Abstract figure legend** Schematic illustration of the bidirectional causative link between cerebral amyloid-beta ($A\beta$) angiopathy and cardiovascular disease in Alzheimer's disease (AD). Common cardiovascular risk factors like microvascular thrombosis, diabetes, atrial fibrillation, hypertension and atherosclerosis lead to cerebral hypoperfusion and progression of AD. Cardiovascular complications in AD arise due to autonomic and endocrine dysregulation, mitochondrial dysfunction and $A\beta$-mediated toxicity to microvasculature and cardiomyocytes.

**Abstract** Alzheimer's disease (AD), the leading cause of dementia, is characterised by cerebral amyloid-beta (A$\beta$) and tau deposition, impairing cognition. While cardiovascular diseases exacerbate AD, the reverse association is underappreciated. This systematic review examined clinical and experimental studies that explored the cardiogenic dementia hypothesis and mechanisms by which amyloidosis in AD contributes to cardiovascular complications. A review of PubMed, Ovid Embase/Medline, and CINAHL conducted in August 2024 identified 252 studies meeting the selection criteria. Evidence links cerebral hypoperfusion from cardiac arrest, heart failure, or orthostatic hypotension to AD pathology, while atherosclerosis and hypertension drive neurodegeneration and cerebral amyloidosis. Vascular scoring tools, such as the Framingham Risk Score, may predict an individual's risk of cognitive impairment. Cardiac amyloidosis correlated with ECG abnormalities, aortic valve calcification, cardiomyopathy and atrial fibrillation. A$\beta$ peptides and AD-related genes exacerbate cardiac fibrosis, negative inotropy and heart rate changes, reduce nitric oxide-mediated vasodilatation, and increase oxidative stress. Preclinical studies revealed that $\beta$-secretase impacts cardiac repolarisation by interfering with delayed rectifier current, although clinical evidence for arrhythmogenesis remains conflicting. AD-related autonomic dysregulation, particularly parasympathetic dysfunction, predisposes to arrhythmias. Additionally, hypercortisolaemia observed in AD has been associated with increased arterial stiffness. Diminished melatonin levels in AD were also linked to endothelial and mitochondrial dysfunction. This review enhances our understanding of how cerebral and cardiac amyloidosis, autonomic dysfunction, and endocrinopathy contribute to cardiac complications in AD, paving the way for research into targeted therapies.

(Received 1 December 2024; accepted after revision 2 April 2025; first published online 30 April 2025)

**Corresponding author** Mastitskaya Svetlana: Department of Translational Health Sciences, Bristol Medical School, University of Bristol, Bristol, UK. Email: svetlana.mastitskaya@bristol.ac.uk

## Introduction

Dementia is a syndrome that causes deterioration in one's cognitive function and ability to carry out the activities of daily life (Varadharajan et al., 2023). Globally, the most common cause of dementia is Alzheimer's disease (AD), accounting for up to 80% of cases (Chen, He et al., 2023). Currently, AD affects over 55 million people, with over 150 million people predicted to be affected by 2050 (Varadharajan et al., 2023). Clinically, AD presents as an insidious impairment to a patient's memory followed by irreversible impairment to orientation, problem-solving and language skills (Schifilliti et al., 2010; Yu & Zhong, 2018). Inevitably, AD causes death from complications such as pneumonia and starvation (Thakral et al., 2023). Unfortunately, no cure exists

(Cattaneo & Capsoni, 2019), with a drug failure rate in clinical trials of 99.6% (Manyevitch et al., 2018), implying a great need for new therapeutics.

While its exact cause and pathophysiology remains unknown (Rezaul Islam et al., 2024), numerous hypotheses exist regarding the development of AD. Traditionally, AD was attributed to the loss of cholinergic neurons with depletion of acetylcholine in the cerebral cortex (Maltsev et al., 2011). Subsequently, the amyloid and tau hypotheses were introduced delineating the roles of amyloid-beta (A$\beta$) and hyperphosphorylated tau (p-tau) proteins, respectively (Maltsev et al., 2011). AD is the result of cerebral amyloidosis, a process whereby misfolded protein fibrils are deposited in the extracellular spaces of tissues (Bravo & Dorbala, 2017; D'Aguanno et al., 2020). Histopathological hallmarks of AD include

**Samuel Parker** is due to graduate with an MBChB from Lancaster University, UK, in 2025. His interest in cardiology inspired him to complete a master's degree in Translational Cardiovascular Medicine at the University of Bristol, UK, in 2024. His master's dissertation explored the brain–heart axis in Alzheimer's disease, specifically, how cerebral amyloidosis affects cardiovascular physiology, and is his first published paper as first author. His future aspirations are to become an academic cardiologist, with a research interest in the brain–heart axis and electrophysiology.

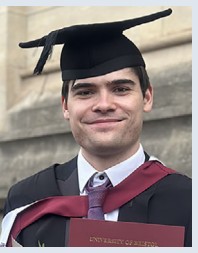

Aβ plaques, which diminish acetylcholine synthesis and acetylcholine nicotinic receptor function (Tran et al., 2002), and neurofibrillary tangles (NFTs) consisting of p-tau fibrils (Janeiro et al., 2022; Yubolphan et al., 2024). NFTs damage the neuronal microtubules required for neuronal function, mitosis and protein transport (Dave, Shah et al., 2023). The precursor of Aβ, amyloid precursor protein (APP), undergoes proteolytic cleavage via two pathways: an amyloidogenic pathway producing Aβ through the action of β-site APP-cleaving enzyme 1 (BACE1), and a non-amyloidogenic pathway that prevents Aβ accumulation (Cole & Vassar, 2007; Yubolphan et al., 2024), as shown in Fig. 1. Mutations in the APP gene, and the genes presenilin 1 (PS1) and 2 (PS2) can predispose to AD pathogenesis. PS1 and PS2 proteins form the γ-secretase complex, which is involved in the cleavage of APP peptides to form Aβ (Dubey et al., 2020). Mutations in the PS1 and PS2 genes characterise an aggressive form of familial AD (Maltsev et al., 2011) through intensifying this proteolysis pathway to form larger quantities of Aβ40 and Aβ42, which are the two most abundant and toxic Aβ species that form plaques (Dubey et al., 2020; Maltsev et al., 2011). Furthermore, the ε4 allele of the apolipoprotein E (APOE) gene promotes AD development due to increased oxidative stress and reduced Aβ clearance (Dubey et al., 2020).

Histologically, Aβ may deposit within the walls of cerebral blood vessels, termed cerebral amyloid angiopathy (CAA), or the walls of peripheral vessels (Xu, Xu et al., 2023). CAA is present in up to 80% of healthy octogenarians and up to 90% of AD cases (Xu, Xu et al., 2023; Yubolphan et al., 2024). Risk factors for AD and CAA include older age, female sex, genetic susceptibility, depression and cardiovascular disease (CVD) (Babaei, 2021). The brain–heart axis outlines the interaction between AD and CVDs such as coronary heart disease (CHD) and heart failure (Xu, Xu et al., 2023).

As shown in Fig. 2, CAA may be the result of cerebral hypoperfusion in the context of CHD or heart failure (Thong et al., 2023). Another cause of cerebral hypoperfusion is during cardiopulmonary bypass (CPB)-induced cardiac arrest, which may lead to postoperative cognitive dysfunction (POCD) in up to 43% of patients after heart surgery (Scott et al., 2014). Given that the brain utilises up to one-fifth of the body's total oxygen consumption, it is unsurprising that cardiovascular insufficiency may induce Aβ and p-tau deposition (Jabeen et al., 2022) and subsequent cognitive dysfunction, termed cardiogenic dementia (Daniele et al., 2020). On the other hand, recent research has suggested the reverse association. Indeed, it is suggested that CAA-induced brain injury can result in heart failure, myocardial infarction, and arrhythmia (Xu, Xu et al., 2023). Additionally, cerebral amyloidosis may affect autonomic nervous system outputs, resulting in an imbalance between its sympathetic (SNS) and parasympathetic (PNS) components. Consequently, this may predispose to cardiovascular sequelae through effects on the heart (Thong et al., 2023; Wang, Pan et al., 2022; Xu, Xu et al., 2023). Amyloidosis may also cause neuroendocrine dysfunction due to impairment of the hypothalamic–pituitary–adrenal (HPA) axis and

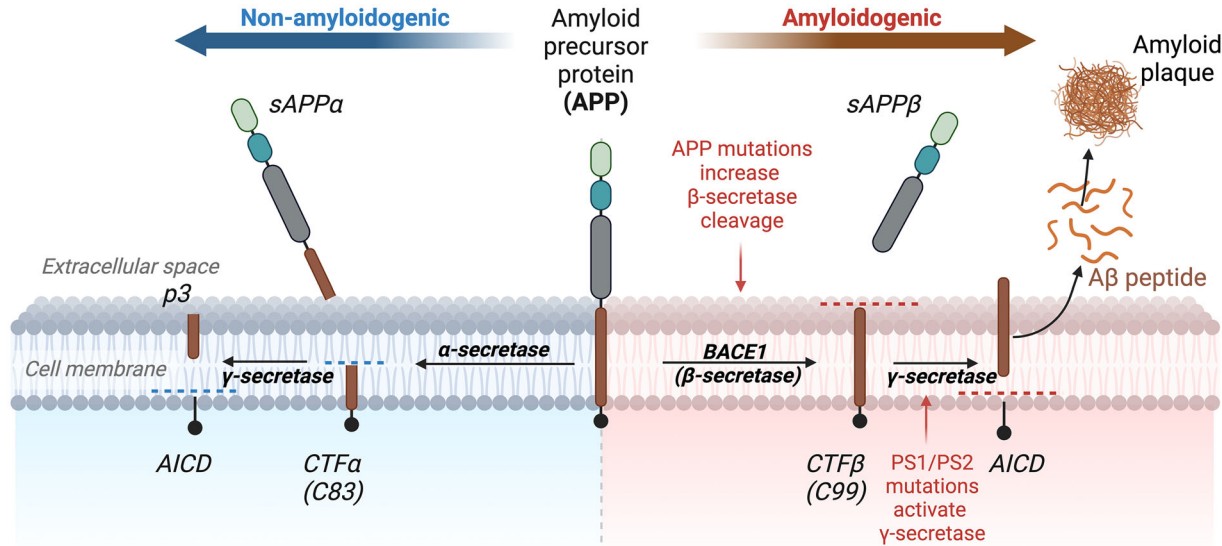

**Figure 1. APP processing pathways**
The amyloidogenic and non-amyloidogenic pathways of amyloid precursor protein (APP) cleavage. A balance between these pathways is crucial in determining Aβ levels, adapted from numerous sources (Cole & Vassar, 2007; Yubolphan et al., 2024). sAPP, soluble amyloid precursor protein; CTF, C-terminal fragment; AICD, APP intracellular domain; BACE1, β-secretase; p3, p3 peptide (a benign product of APP cleavage).

glucocorticoid release into the blood (Xu, Xu et al., 2023); specifically, cortisol in humans and corticosterone in rodents (Dong & Csernansky, 2009).

In addition, systemic amyloidosis may affect myocardial extracellular spaces (Fontana et al., 2019). The majority of cardiac amyloidosis is attributed to misfolded immunoglobulin fibrils produced by plasma cells, called light chain amyloidosis (Fontana et al., 2019), which comprises nearly two-thirds of all amyloidosis cases in the United Kingdom (D'Aguanno et al., 2020). The next most common type is transthyretin amyloidosis (ATTR), due to the deposition of hepatic transthyretin protein, comprising wild-type (ATTRwt) and hereditary (ATTRm) forms (Bravo & Dorbala, 2017; Koike & Katsuno, 2021). Cardiac amyloidosis is associated with high mortality due to cardiomyopathy and heart failure, occurring in nearly all patients with ATTRwt (Bravo & Dorbala, 2017; Koike & Katsuno, 2021).

CVDs are responsible for one-third of deaths globally (Uijl et al., 2016). Therefore, the coexistence of CVDs with AD creates a challenging, comorbid patient population. Understanding the relationship between these pathologies is crucial for the development of novel therapeutics. Accordingly, this systematic review aims to comprehensively summarise the existing knowledge on cardiogenic dementia and describe how amyloidosis can lead to cardiovascular complications in the context of systemic amyloidosis and AD. To our knowledge, this is the first systematic review with this aim, and the first

to cover autonomic and hormonal dysfunction alongside Aβ and tau amyloidosis.

## Objectives

The objectives of this systematic review are to:

(i) explore the cardiogenic dementia hypothesis of AD;
(ii) investigate the impacts of Aβ and tau amyloidosis on cardiac function;
(iii) clarify how cerebral amyloidosis can cause autonomic and hormonal dysregulation and the effects on subsequent CVD.

## Methodology

The search for this review was conducted in August 2024 using PubMed, Embase, MedLine and CINAHL, in accordance with the 2020 Preferred Reporting of Systematic Reviews and Meta-analyses (PRISMA) guidelines (Page et al., 2021). There were no restrictions based on the date of publication. Initially, three separate searches were performed using the following terms:

(i) (cardiac* OR heart) AND ((Amyloid*) AND (Alzheimers OR Alzheimer's))
(ii) ((Amyloid*) AND (Alzheimers OR Alzheimer's)) AND (arrhythm* OR autonomic dysfunction OR

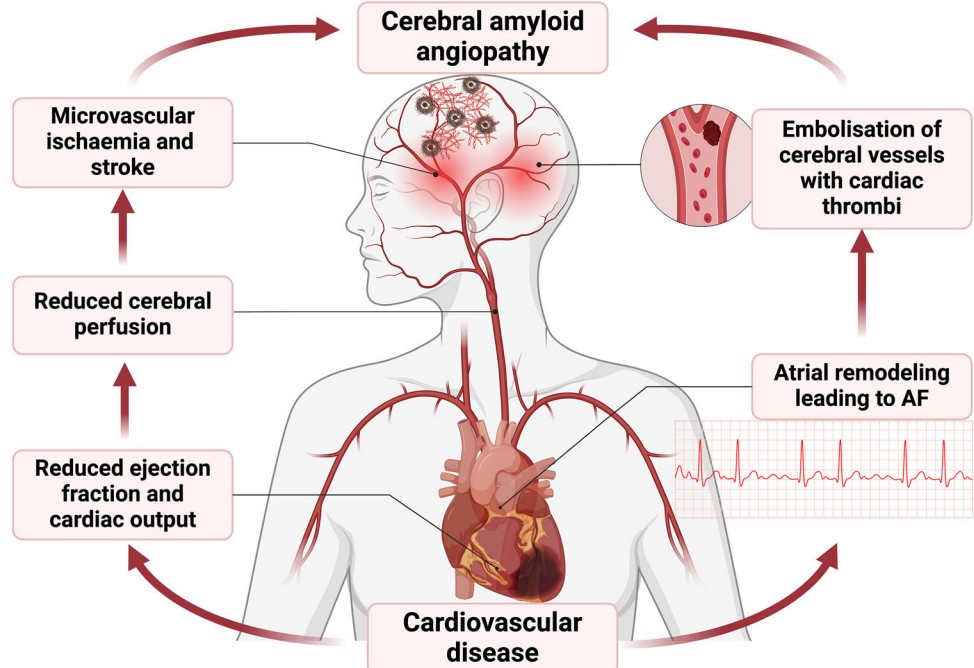

**Figure 2. CVD as a driver of dementia**
A diagram demonstrating how cardiovascular diseases can lead to cerebral amyloid angiopathy and dementia through the brain–heart axis, adapted from Thong et al. (2023). AF, atrial fibrillation.

neuroendocrine dysfunction OR neurohumoral dysfunction)

(iii) (autonomic* OR neurohumoral*) AND ((Amyloid*) AND (Alzheimers OR Alzheimer's)).

Firstly, titles and abstracts were screened for relevance to cardiovascular complications of AD. For this review, non-English papers without a translation, editorials, conference abstracts, case reports, papers not mentioning the heart or peripheral vascular system, papers regarding non-AD dementia, and other review articles were excluded. Next, the full texts of relevant papers were screened and included if they investigated the impact of cardiovascular pathology on the development of AD, or vice versa. Then, the included papers were classified according to the following sections: Cardiogenic Dementia, Amyloid Fibrils within the Heart, Amyloid Beta and the Cardiovascular System, Tau Protein and the Cardiovascular System, AD Genes and Cardiac Physiology, Autonomic Dysregulation in AD, and Hormonal Alterations in AD, as outlined below.

## Results

As demonstrated in Fig. 3, 5783 results were found in total from all three databases: PubMed, Ovid Embase/Medline and CINAHL yielded 2549, 3102 and 132 results, respectively. After title and abstract screening, 3753 studies were excluded across the three database searches based on the exclusion criteria. Consequently, this left 424 papers for detailed screening. Upon full-text screening, 173 were excluded due to irrelevance, no mention of the heart/cardiovascular system, being identified as extra duplicates, or being review articles, leaving 252 papers for analysis, as summarised in Supplementary Table. To avoid repetition, this table will not be directly referred to again in this review but contains details of all cited studies. Some papers may pertain to more than one section and have been referenced accordingly throughout the review.

**Cardiogenic dementia.** Studies identified that pertain to 'cardiogenic dementia' were categorised into separate themes including cardiac arrest, CPB, heart failure, orthostatic hypotension, atherosclerosis, hypertension, vascular risk scores, platelet activation and thrombosis, and arterial stiffness.

*Cardiac arrest increases vulnerability to Alzheimer's disease.* Three clinical studies (Ashton et al., 2023; Wiśniewski & Maślińska, 1996; Zetterberg et al., 2011) and one preclinical study (Kocki et al., 2015) were found showing how cardiac arrest may predispose to AD pathology through plaque deposition and amyloidogenic processing. Observational studies provided evidence of

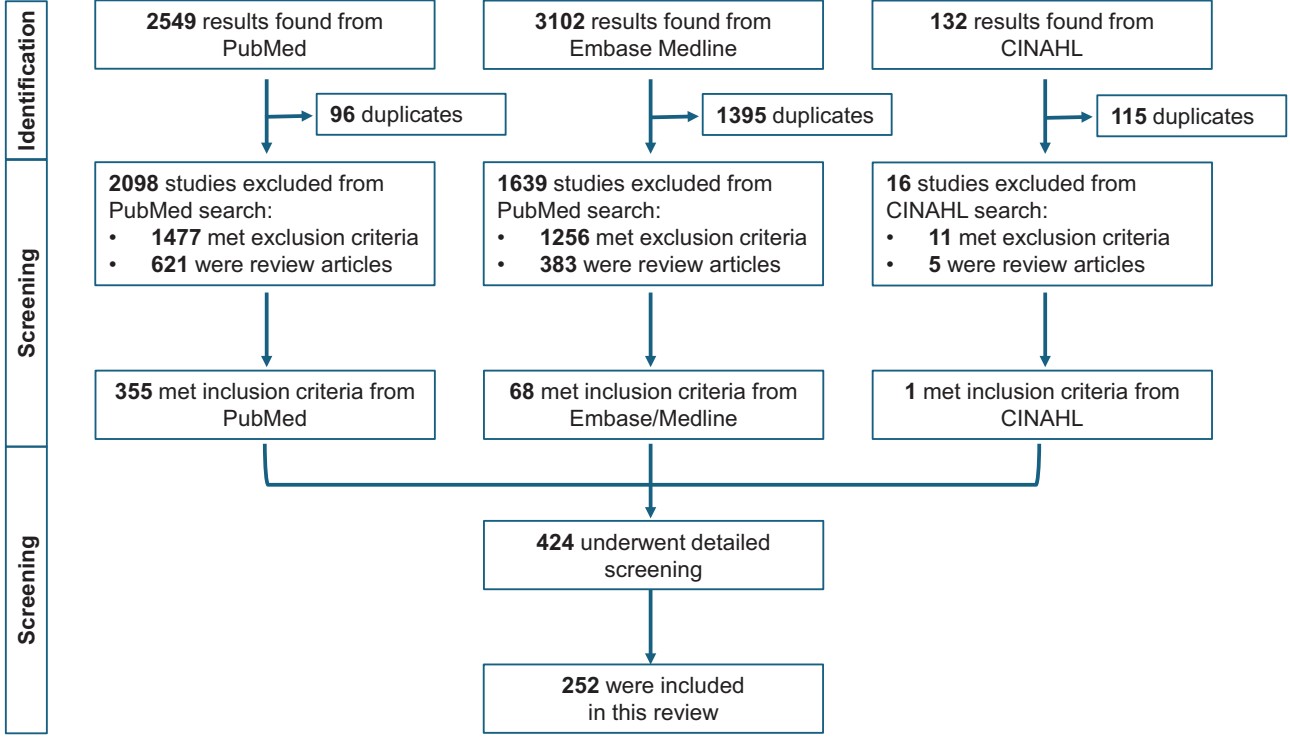

**Figure 3. PRISMA flow diagram**
A flowchart outlining the inclusion process for studies in this review, in accordance with PRISMA guidelines. The databases were screened sequentially in the following order: PubMed, Embase/Medline and then CINAHL.

elevated p-tau, A$\beta$40 and A$\beta$42 after cardiac arrest, which correlated to poor neurological function (Ashton et al., 2023; Wiśniewski & Maślińska, 1996; Zetterberg et al., 2011). Furthermore, Kocki et al. (2015) noted increases in hippocampal AD gene expression after cardiac arrest in rats (Kocki et al., 2015).

*Cardiopulmonary bypass may induce neuronal injury and cerebral amyloidosis.* Nine articles demonstrated the effect of cardiac surgery and CPB on AD pathology (Alifier et al., 2020; Evered et al., 2009; Hu et al., 2016; Klinger et al., 2018; Palotás et al., 2010; Požgain et al., 2022; Reinsfelt et al., 2013; Sparks et al., 2000; Wang, Cao et al., 2022). Rates of POCD after cardiac surgery ranged from 13% to 57% (Evered et al., 2009; Klinger et al., 2018; Wang, Cao et al., 2022). Cross-sectional analysis in patients undergoing CPB showed that CSF A$\beta$42 and tau protein levels increase after cardiac surgery (Alifier et al., 2020; Palotás et al., 2010; Požgain et al., 2022; Reinsfelt et al., 2013) alongside markers of neuronal injury (Alifier et al., 2020; Palotás et al., 2010), and this predisposed to poorer cognition post-surgery (Požgain et al., 2022). While some studies show that A$\beta$40 and A$\beta$42 peptides decrease after CPB (Evered et al., 2009; Hu et al., 2016; Palotás et al., 2010; Wang, Cao et al., 2022), and this is what increases the risk of cognitive impairment (Evered et al., 2009), this could indicate an increase in cerebral amyloidosis and hence lower circulating A$\beta$ levels. However, some publications show how cerebral amyloidosis is unaffected by CPB and does not predispose to POCD (Klinger et al., 2018; Wang, Cao et al., 2022). One study in pigs by Sparks & colleagues (2000) showed increased cerebral A$\beta$-positive neurons after CPB but the quality of cardio-protection with CPB did not affect A$\beta$ staining (Sparks et al., 2000). As shown, studies agree that CPB generates neuronal injury, but they disagree on the effect of CPB on CSF and plasma levels of AD biomarkers and their relationship to cognitive dysfunction.

*Reduced ejection fraction in heart failure increases cerebral amyloidosis.* The study by Sparks et al. (2000) reported that reduced cardiac output in pigs does not increase cerebral A$\beta$ deposition (Sparks et al., 2000). Animal studies agree that heart failure increases cerebral APP peptide expression and BACE1 activity but disagree on the effect of heart failure on A$\beta$ and tau protein levels (Baranowski et al., 2021; Hong et al., 2013). Of note, one study suggested that the effects of heart failure on cerebral BACE1 and APP gene transcription and cognitive dysfunction were more severe in female mice than male mice (Hong et al., 2013).

Regarding subclinical AD, Santos et al. (2016) showed that rate pressure product (the mathematical product of systolic blood pressure and heart rate, a proxy for myocardial oxygen consumption and workload) was moderately associated with cerebral amyloidosis and cognitive impairment in patients without CVD (Santos et al., 2016). Similarly, a sub-analysis of the Atherosclerosis Risk in Communities (ARIC) study, known as the ARIC-PET study, discovered age-dependent effects of end-diastolic left ventricle diameter on cerebral A$\beta$ deposition, demonstrated by a higher standardised uptake value ratio on cerebral imaging (Johansen et al., 2019). A separate analysis of patients from the ARIC study showed that numerous proteins involved in CVD including natriuretic peptide precursor B (NPPB), a protein expressed by the myocardium and a known hormonal biomarker of heart failure, associated with the development of dementia (Tin et al., 2023). Another protein that is elevated when there is atrial or ventricular overload in heart failure, called N-terminal pro-BNP, was found to be associated with an increased risk of dementia in the Whitehall II and ARIC studies (Lindbohm et al., 2022). Likewise, the CABLE study showed that a lower left ventricular ejection fraction (LVEF) and greater size of the left heart chambers correlated to higher CSF p-tau and poorer cognition (Hu et al., 2024; Zheng et al., 2021), but did not assess amyloidosis in the brain parenchyma. Similarly, an analysis of patients from the Amsterdam Dementia Cohort found that a reduction in cerebral blood flow increases the cerebral amyloid burden, and vice versa (Ebenau et al., 2023). Trieu et al. (2024) recruited patients from the Heart–Brain study who had heart failure, carotid artery occlusion and vascular cognitive impairment, or who were healthy controls. Interestingly, they discovered that p-tau and glial fibrillary acidic protein, a marker of neuro-inflammation in AD, associated with baseline cognitive impairment, but not cognitive decline over time in heart failure (Trieu et al., 2024), and limited associations between carotid artery disease and AD biomarkers were noted (Trieu et al., 2024). Aside from A$\beta$ and tau peptides, a murine study showed that heart failure is associated with a rise in levels of BACE1 protein in the cortex and hippocampus (Nural-Guvener et al., 2013).

In contrast, other authors showed that lower LVEF only increases CSF tau biomarkers in patients without cognitive impairment and, counterintuitively, that an increased LVEF associated with greater levels of CSF A$\beta$42 in females, and that no association existed in males (Kresge et al., 2020). Moreover, a sub-analysis of the PARAGON-HF study by Dewan et al. (2024) revealed that a high proportion of patients with heart failure with a preserved LVEF had poor cognitive performance and that pharmacological therapy did not improve cognition (Dewan et al., 2024). These findings demonstrate that heart failure may affect AD pathology in a sex-specific manner, where females with poorer LVEF exhibit more severe AD neuropathology.

Several studies suggested mechanisms for how a reduction in LVEF, and subsequent hypoperfusion of the brain parenchyma, may predispose to A$\beta$ amyloidosis.

Kaufman & colleagues (2021) suggested that this may be mediated by the APOE-$\varepsilon$4 allele (Kaufman et al., 2021). Alternatively, Miners et al. (2018) discovered associations between hypoperfusion of the brain parenchyma and blood–brain barrier breakdown as well as loss of pericytes, and how this correlates to cerebral A$\beta$ plaque burden (Miners et al., 2018). It is possible that pericyte loss may occur not only in the brain but also the hearts of patients with AD. Regardless of the mechanism, the research indicates that heart failure can predispose to greater amyloidogenic processing as well as increased concentrations of A$\beta$ and tau peptides.

*Orthostatic hypotension increases susceptibility to dementia and Alzheimer's biomarker changes.* Two works described orthostatic hypotension in the development of dementia (Ruiz Barrio et al., 2023; Zhang et al., 2021). These studies describe that orthostatic hypotension and subsequent cerebral hypoperfusion as a result of diabetic neuropathy and baroreceptor reflex dysfunction predispose to increased AD biomarkers, especially p-tau (Zhang et al., 2021), but that orthostatic hypotension due to Parkinson's disease does not affect AD biomarkers (Ruiz Barrio et al., 2023). Thus, the cause of orthostatic hypotension may determine the subsequent effects of AD amyloidosis.

*Atherosclerosis and its risk factors precede Alzheimer's disease neuropathology.* The association between CVD and AD is observed in animal models, where diabetic mice exhibit more severe CAA (Vargas-Soria et al., 2022). Numerous publications suggest that CVDs may correlate to increased cortical atrophy (de Silva et al., 2022; Vemuri et al., 2017; Vescio & Pattini, 2024), lower plasma A$\beta$42 levels and A$\beta$42:A$\beta$40 ratio (each indicating a greater risk of AD) (Seppälä et al., 2010) and greater cerebral A$\beta$ plaque deposition (Liu et al., 2019; Sparks, 1997). In a study by Twait et al. (2024), investigating patients with evidence of arterial disease, markers of axonal injury and neuroinflammation were associated with cerebral infarction and neurodegeneration (Twait et al., 2024). The CABLE study provided evidence of greater tauopathy in individuals with CVD (Li et al., 2024). Investigating the link between atherosclerosis and AD, Baradaran et al. (2022) showed that participants from the Framingham offspring cohort with increased carotid intima-media thickness, a marker of carotid atherosclerosis, exhibited greater cerebral A$\beta$ plaque deposition (Baradaran et al., 2022). A German–Dutch memory clinic study also produced similar results (Kučikienė et al., 2022). Likewise, analyses of patients from the Rotterdam study cohort demonstrate that A$\beta$40 associated with coronary artery calcification (Wolters et al., 2022) and that arterial calcification interacts with A$\beta$42 to predispose to poorer cognitive performance, but cardio-metabolic risk factors mediated this association (Frentz et al., 2024). This may

provide further evidence of a role for extra-cerebral atherosclerosis in the development of AD.

To explore the role of atherosclerosis in AD further, Shabir et al. (2022) induced atherosclerosis in mice to create a model of atherosclerosis and created another murine model of atherosclerosis superimposed on AD (Shabir et al., 2022). They discovered that the atherosclerotic models showed reduced cerebral haemodynamic regulation and reduced concentrations of oxygenated haemoglobin. By measuring vascular reactivity to hypercapnia, it was shown that these results were due to atherosclerosis in the extra-cerebral arteries. Additionally, the mixed models developed more hippocampal A$\beta$ plaques than the other murine models, and the model of atherosclerosis showed upregulation of tumour necrosis factor-$\alpha$ and interleukin-1$\beta$. Moreover, diseased mice showed greater cortical spreading depression, which may indicate greater cerebral ischaemia (Shabir et al., 2022). As the reactivity of the cerebral arteries in response to hypercapnia was preserved in all the models, this suggests that extra-cerebral atherosclerosis promotes increased cerebral A$\beta$ amyloidosis and neuro-inflammation, which is in agreement with a murine study by Zhang & colleagues (2020) (Zhang & Luo, 2020). Supporting this, Cheng et al. (2021) concluded that neurodegeneration, A$\beta$ plaque deposition and cognition post-myocardial infarction are dependent on the activation of microglial PYD domains containing protein 3 (NLRP3) in the brain, which is important for assembling the interleukin-1$\beta$-producing inflammasome (Cheng et al., 2021). Therefore, neuroinflammation may be a key mediator of AD after CVDs such as myocardial infarction. Intriguingly, an autopsy study using a cohort from the Centre for Neurodegenerative Disease Research showed that while vascular risk factors were associated with cerebral infarction, and patients with AD tended to have more vascular risk factors than controls, there was no association with CAA (Robinson et al., 2022). Potentially, systemic CVDs may predispose to pathology in the brain parenchyma rather than angiopathy.

Conversely, there is substantial research to suggest that CVDs may not precede the development of AD pathology. Specifically, CVDs may not increase the risk of cerebral A$\beta$ or tau amyloidosis (Kosunen et al., 1995; Vemuri et al., 2017). In agreement with this, the study mentioned above by Trieu et al. (2024) showed that the A$\beta$42:A$\beta$40 ratio, where a lower ratio indicates greater cerebral A$\beta$ deposition, and levels of p-tau, did not differ between patients with CVDs and healthy controls (Trieu et al., 2024). Indeed, Alafuzoff & Libard (2020) performed postmortem immunohistochemistry on subjects with and without cognitive impairment and found that histological evidence of CVD did not correspond to AD-like pathology (Alafuzoff & Libard, 2020). Also, increased left ventricle (LV) mass index (indicating

ventricular remodelling, a potential consequence of hypertension and systemic CVD) was not shown to predispose to increased AD biomarkers (Moore et al., 2022) and ventricular hypertrophy due to angiotensin II-induced hypertension had no effect on cerebral $A\beta$ amyloidosis in two models of AD (Hendrickx et al., 2022). Concordantly, Smith et al. (2018) showed no effect of hypertension on cerebral $A\beta$ uptake (Smith et al., 2018). Moreover, analysis of patients from the ARIC study by Lu et al. (2024) showed that CHD did not predispose to changes in the $A\beta42:A\beta40$ ratio or markers of axonal injury (Lu et al., 2024). However, the presence of coronary artery disease in both human subjects and rabbits correlated with greater activation of hippocampal microglia and $A\beta$ amyloidosis (Streit & Sparks, 1997), which could indicate greater neuroinflammation and thus development of AD pathology.

*Hypertension increases Alzheimer's disease biomarkers, neurodegeneration and cognitive decline.* Many studies indicate that hypertension, a prominent cardiovascular risk factor, also predisposes to AD neuropathological changes. Lu et al. (2024) showed that hypertension and diabetes are associated with biomarker changes indicative of AD (Lu et al., 2024). Regarding mechanisms for this, murine studies conclude that hypertension predisposes to dysregulation of certain hippocampal genes involved in AD (Csiszar et al., 2013), cerebral $A\beta$ amyloidosis (Gentile et al., 2009; Kruyer et al., 2015; Lai et al., 2021) and neurodegeneration (Kruyer et al., 2015), although one study showed no altered $A\beta$ processing and parenchymal $A\beta$ deposition despite cerebrovascular damage and loss of cortical myelin due to hypertension (Lai et al., 2021). Nevertheless, an autopsy analysis by Richardson & colleagues (2012) concluded that hypertension associated with greater microinfarction and CAA (Richardson et al., 2012). Stemming from this, an analysis of Insight 46 (a neuroimaging study aiming to identify brain changes associated with healthy ageing and a sub-study of the National Survey of Health and Development) showed that a greater blood pressure increases white matter hyperintensities on magnetic resonance imaging, and provides evidence to suggest that hypertension may increase neurodegeneration in later life (Lane et al., 2019). Another mechanism for how hypertension may predispose to AD and cognitive impairment may be through increased tauopathy (Hu et al., 2022), and there is evidence that hypertension may lower the plasma $A\beta42:A\beta40$ ratio, perhaps indicating exacerbated cerebral $A\beta$ amyloidosis (She et al., 2021). This is concordant with the Swedish BioFINDER study (Janelidze et al., 2016), the Honolulu-Asia Aging study (Petrovitch et al., 2000; Shah et al., 2012) and The Tobago Health Study (Rosano et al., 2024). The presence of hypertension may also translate into clinically significant cognitive impairment, as

found by the DIAN study (Xu, Aung et al., 2023). Indeed, diabetes mellitus has also been shown to potentiate the risk of AD in individuals with hypertension (Ruthirakuhan et al., 2024).

A study among rural-dwelling Chinese adults by Ren et al. (2024) suggested that hypertension and other CVDs do not increase the risk of AD (Ren et al., 2024). This is in agreement with results of a study using findings from the National Alzheimer's Coordinating Centre Database in the United States, which showed a negative association between hypertension and AD neuropathology (Scambray et al., 2023). Furthermore, the literature suggests that lower blood pressure may mediate the relationship between atherosclerosis and hippocampal volume (Kapasi et al., 2024), and that hypertension may lower the odds of AD (Tosto et al., 2016), although hypertension may indirectly increase the risk of AD by increasing the risk of having a stroke (Tosto et al., 2016). In addition, an analysis of participants from the Framingham Heart study showed that increased plasma $A\beta42$ associated with a reduced risk of AD and dementia (Chouraki et al., 2015). Consequently, the above association of hypertension with $A\beta$ (Janelidze et al., 2016) may not translate to a greater risk of clinical dementia syndrome.

*Vascular risk scores may be useful in predicting the development of Alzheimer's disease.* Further evidence for the link between AD and CVD can be obtained through assessment of an individual's general cardiovascular risk. For example, higher concentrations of low-density lipoprotein cholesterol, a risk factor for AD and vascular disease, may modulate the interaction between deposition of $A\beta42$ and tau peptides (Han et al., 2024; Kuo et al., 1998), although other factors, such as age and educational attainment, may confound this association (Bennett et al., 2020). Additionally, the Multi-Ethnic Study of Atherosclerosis showed how subclinical factors pertaining to blood pressure, atherosclerosis and cardiac function reduce the time to onset of dementia (Hughes et al., 2024) and correlate to greater neurodegeneration and white matter hyperintensity volumes (Lockhart et al., 2022). Indeed, vascular risk factors predispose to greater cerebral $A\beta$ plaque amyloidosis in APOE-$\varepsilon4$ carriers (Sapkota et al., 2023), and Kim et al. (2019) provided evidence that the presence of vascular risk factors mediates the association between imaging evidence of $A\beta$ amyloidosis and subjective cognitive impairment (Kim et al., 2019). Indeed, other authors showed that vascular risk factors act alongside the presence of preclinical AD to increase neurofilament light concentrations over time (a neuronal cytoskeletal protein whose levels in the blood and CSF indicate neurodegeneration and axonal damage), but they concluded that vascular risk factor burden does

not associate with AD pathophysiology or biomarkers (Ferrari-Souza et al., 2024).

Furthermore, García-Lluch et al. (2024) recruited patients and controls aged between 50 and 75 to assess the use of cardiovascular risk scoring systems to predict the presence of AD (García-Lluch et al., 2024). Greater ERICE (a cardiovascular predictive score for Spanish population) and SCORE2 (an algorithm predicting 10-year risk of CVD onset in European populations) values in this study were associated with the presence of AD, greater levels of CSF t-tau and neurofilament light, and lower $A\beta42:A\beta40$ ratios (García-Lluch et al., 2024). Indeed, Shirzadi et al. (2024) utilised linear regression with imaging evidence of $A\beta$ and tau deposition, cerebral blood flow, Framingham Heart study risk (an estimation of a 10-year risk of CVD onset in non-symptomatic patients) and white matter hyperintensity as the variables to explain cognitive decline in AD. Cardiovascular risk factors and cerebral blood flow independently associated with cognitive decline, measured by the Preclinical Alzheimer Cognitive Composite (PACC), which includes scores from various cognitive assessments (Shirzadi et al., 2024). Intriguingly, findings by García-Lluch et al. (2024) and Bilgel et al. (2021) suggest that the Framingham Risk Score (FRS) does not associate with AD pathology (Bilgel et al., 2021; García-Lluch et al., 2024). The FRS includes data regarding age, sex, blood pressure, cholesterol, diabetes and smoking to predict an individual's 10-year risk of CVD (Bilgel et al., 2021). In addition, Rabin et al. (2018 and 2022) discovered that vascular risk illustrated by the FRS is not associated with cerebral $A\beta$ burden, but these two parameters interacted to accelerate cortical atrophy, and a higher vascular risk was associated with cognitive decline measured by PACC (Rabin et al., 2018; Rabin et al., 2022). Perhaps, an individual's FRS interacts with age to predispose to neuropathology in AD, which is suggested in a paper by Conner & colleagues (2019) (Conner et al., 2019). Although, an abundance of research suggests that the FRS may directly increase AD biomarkers (Jiang et al., 2023), cognitive decline (Yu et al., 2021), neurodegeneration (Tranfa et al., 2024) and cerebral white matter hyperintensities (small lesions and microbleeds detected on brain scans), which are associated with AD (Pålhaugen et al., 2021; Saeed et al., 2024). Additionally, a higher FRS interacts with cerebral $A\beta$ burden to accelerate cerebral tau amyloidosis in patients without dementia (Rabin et al., 2019; Yau et al., 2022). Likewise, a sub-analysis of the 'Insight 46' study showed that a higher FRS in older patients associate with poor white matter integrity, and that midlife hypertension leads to poor white matter integrity in female patients only (James et al., 2023) and accelerated hippocampal and whole-brain atrophy (Keuss et al., 2022). One potential mechanism of how vascular risk factors predispose to AD was explored by Lin & colleagues (2021) (Lin et al., 2021). They observed increased blood–brain barrier permeability in participants with higher vascular risk, determined by the presence of hypertension, diabetes, smoking and body mass index. Participants with blood–brain barrier breakdown demonstrated poorer cognition and a lower CSF $A\beta42:A\beta40$ ratio (Lin et al., 2021). Therefore, vascular disease may predispose to AD through breakdown of the blood–brain barrier. However, it is worth noting that these authors used a composite vascular risk score using variables including five cardiovascular measures to assess vascular risk (Lin et al., 2021), not the FRS.

Contradicting the papers above, several articles provide evidence against an association between FRS and AD. The trend between FRS and cerebral atrophy discovered by Insight 46 was only noted in patients without $A\beta$ amyloidosis; patients with imaging evidence of cerebral $A\beta$ plaques did not demonstrate this association (Keuss et al., 2022). Indeed, it has been shown that cardiovascular risk does not independently associate with $A\beta$ or tau amyloidosis (Rabin et al., 2019; Saeed et al., 2024). One autopsy study showed that overall FRS did not associate with a diagnosis of AD or its biomarkers, and patients with fewer vascular risk factors, demonstrated using the FRS, showed a greater association between cerebral $A\beta$ and tau burden (Oveisgharan et al., 2020).

Despite these contradictory findings, the evidence supporting an association between FRS and AD pathology strengthens the cardiogenic dementia hypothesis. Furthermore, the evidence suggests that numerous vascular risk scores could be useful in predicting an individual's risk of AD.

*Platelet activation and thrombosis impact Alzheimer's biomarker changes.* Further evidence for the cardiogenic dementia hypothesis comes from articles investigating thrombosis and hypoxia due to cardiovascular pathology. Wolska et al. (2023) suggested a role for platelets during cerebral amyloidosis in AD, which may suggest a further link between hypoxia during CVDs, where thrombosis can limit organ perfusion during a myocardial infarction, for example, and AD pathogenesis (Wolska et al., 2023). An *in vitro* study by Wolozin et al. (1998) highlights that increased platelet aggregation due to $A\beta40$ may require adenosine diphosphate and fibrinogen (Wolozin et al., 1998). Developing this hypothesis, increased platelet aggregation in response to adenosine diphosphate associated with increased risk of AD after 20 years in an *in vitro* analysis of platelets from patients in the Framingham Heart study (Ramos-Cejudo et al., 2022). A study by Abubaker et al. (2019) corroborated this, showing that platelet adhesion *in vitro* increased in response to $A\beta42$ peptides at venous shear stress level, but not at arterial level (Abubaker et al., 2019), which implies that this platelet adhesion may not be clinically relevant to human atherosclerosis. This study explored a potential

mechanism for increased platelet activity in response to A$\beta$42, involving activation of the $\alpha_{\text{IIb}}\beta_3$ integrin receptor and thus intracellular signalling pathways (Abubaker et al., 2019). Similarly, Donner et al. (2020) showed that A$\beta$40 can activate the platelet glycoprotein VI and $\alpha_{\text{IIb}}\beta_3$ receptor and increase platelet aggregation. These findings were confirmed by the authors using *in vivo* mouse models (Donner et al., 2020), although one murine study suggests male mice with AD pathology may exhibit protection against thrombosis and its consequences (Donner et al., 2024). Intriguingly, Visconte & colleagues (2018) confirmed that A$\beta$42, A$\beta$40 and A$\beta$25-35 increase platelet adhesion, and while APP peptides did not bind to the $\alpha_{\text{IIb}}\beta_3$ integrin receptor or affect thrombosis, the response of platelets to A$\beta$ peptides was abolished in APP knockout mice (Visconte et al., 2018), suggesting that APP selectively mediates platelet adhesion to A$\beta$ and is responsible for A$\beta$-promoted potentiation of thrombus formation. Nevertheless, the increased aggregation and thrombosis may contribute to hypoxia and cerebral or cardiac hypoperfusion. One study linking hypoxia to AD used participants from the CABLE study (a large-scale cohort study in China aiming to explore genetic and environmental factors influencing AD), and showed that patients with more severe anaemia, and thus decreased oxygen-carrying capacity of red blood cells, had lower levels of CSF A$\beta$42 independent of other variables, implying greater cerebral deposition (Yang et al., 2021). However, this is contradicted by another cross-sectional study (Kim et al., 2021). Overall, these articles suggest that platelet activation, thrombosis and hypoxia may increase AD neuropathology.

*Greater arterial stiffness associates with cerebral amyloid angiopathy and Alzheimer's pathology.*　　The final group of articles evidencing the cardiogenic dementia hypothesis relate to arterial stiffness. Moore et al. (2021) measured aortic stiffness using pulse wave velocity in patients over 73 years old with concomitant CVD and showed that greater aortic stiffness associates with increased CSF p-tau and t-tau (Moore et al., 2021). Although Pasha et al. (2020) concluded that pulse wave velocity does not independently associate with cerebral A$\beta$ deposition (Pasha et al., 2020), they and other authors (Cui et al., 2018; Hughes et al., 2018) provide evidence that greater arterial stiffness increases AD pathology. In addition, evidence suggests that these biomarker changes may translate into clinical dementia in patients, but not specifically AD (Cui et al., 2018). When investigating the proteins involved in arterial stiffening, Wagner et al. (2022) showed that medin, the most common amyloid in humans which aggregates in arteries resulting in stiffening, correlates to greater total levels of A$\beta$ within the arteries (Wagner et al., 2022). In this study, postmortem specimens of cerebral blood vessels from patients with

AD showed co-accumulation of medin and A$\beta$ (Wagner et al., 2022). Additionally, medin has been shown to have a similar primary and secondary structure to A$\beta$ peptides and forms fibrils of similar structure (Davies et al., 2014; Davies et al., 2015). The presence of MFG-E8, the precursor protein cleaved to form medin, associated with a greater decline in cognitive functioning in patients with AD. This suggests a potential association between AD, CAA and arterial stiffening (Wagner et al., 2022).

On the other hand, it has been shown that patients with higher arterial stiffness have poorer cognition despite lower CSF levels of tau (Kumar et al., 2020). In addition, a study in patients from the Framingham Heart study showed that while certain measures of aortic stiffness and pulsatility associated with greater tau amyloidosis, a higher carotid-femoral pulse wave velocity did not associate with increased tauopathy and these parameters had no effect on cerebral A$\beta$ deposition (Cooper et al., 2022). The above evidence suggests a mixed effect of arterial stiffness on AD pathogenesis, but there is more evidence to suggest a pro-Alzheimer's phenotype in patients with greater arterial stiffness.

Taken together, the studies cited in this section suggest that CVDs and their risk factors predispose to cortical atrophy, cognitive decline, CAA and parenchymal amyloidosis. However, there are mixed results regarding the effect of CVDs on AD biomarkers, and studies appear to disagree on the utility of the FRS in predicting the development of AD.

### Amyloid fibrils within in the heart

*Cardiac amyloidosis and the association with Alzheimer's disease.* Most studies that investigated cardiac amyloidosis largely pertained to transthyretin (ATTR) and the formation of pre-amyloid oligomers. Annamalai & colleagues (2017) illustrated cardiac amyloid fibril formation through detection of amyloid characteristics such as green birefringence on Congo red staining, and that fibrils in different locations in the body shared similar structures (Annamalai et al., 2017). Indeed, Nguyen et al. (2024) demonstrated that ATTR fibrils in one patient carrying the V30M mutation in transthyretin had similar structure in the heart and nerves (Nguyen, Afrin et al., 2024). The V30M mutation gives rise to a phenotype of neuropathy and cardiomyopathy, and has been shown by other cross-sectional studies to predispose to cerebral and cerebellar amyloidosis, but to a lesser extent than the amyloidosis in patients with AD (Uneus et al., 2022). As explored later, this provides evidence that amyloidosis can affect both components of the brain–heart axis. The same research group further demonstrated that the I84S and V30M mutations seen in ATTRm result in inter-patient variation in amyloid fibrillary structure, despite similar symptoms (Nguyen, Singh, Afrin, Yakubovska

et al., 2024). Alternatively, ATTRwt fibrils were shown to have consistent structures between patients with cardiomyopathies (Nguyen, Singh, Afrin, Singh et al., 2024), and similar fibril structure has been noted in patients carrying the T60A mutation in transthyretin (Fernandez-Ramirez et al., 2024). In addition, Levites et al. (2024) performed proteomics on human AD brain specimens and mouse models of AD (Levites et al., 2024). They concluded that two proteins which promote cerebral A$\beta$ amyloidosis, pleiotrophin and midkine, are located alongside transthyretin in cardiac amyloidosis (Levites et al., 2024). In another study by King & Robinson (2020), mice with AD pathology exhibited upregulation of numerous proteins involved in protein synthesis and mitochondrial dysfunction in the heart (King & Robinson, 2020). These studies demonstrate the involvement of the brain–heart axis in AD pathology in mice and humans, and how cerebral amyloidosis may be linked to cardiac ATTR.

Larger observational studies provide further evidence for the cardiac effects of transthyretin deposition. In participants from both phases of the Dallas Heart study, the presence of V122I *TTR* alleles, the most common cause of ATTRm globally, was associated with reduced circulating levels of transthyretin (Hendren et al., 2024), which implies cardiac deposition. Furthermore, the V122I mutation predisposed to altered cardiac function and morphology, and transthyretin levels positively correlated with limb lead QRS voltage on electrocardiography (Hendren et al., 2024), suggesting an effect on cardiac electrophysiology. Developing this conclusion, the Hisayama study found that ATTR correlates with heart failure and cerebral amyloidosis (Hamasaki et al., 2022). With regard to aortic valve disease, immunohistochemistry analysis of aortic valve specimens after surgery showed that calcified aortic valves stain positive for transthyretin, APP and A$\beta$ whereas non-calcified areas and healthy valves do not stain for these proteins. This provides evidence for how cardiac amyloidosis, and perhaps AD, may predispose to valvular dysfunction and stenosis (Heuschkel et al., 2020).

Although not relating to ATTR fibrils, Mielcarek et al. (2014) demonstrated electrophysiological abnormalities in a mouse model of Huntington's disease, involving amyloidosis of huntingtin fibrils. Indeed, these mice displayed dislocation of connexin-43 proteins, indicating gap junction disruption due to amyloidosis. Furthermore, there was upregulation of cardiomyopathy-related genes, such as *Nppa* (the gene for atrial natriuretic peptide, ANP), hypertrophy, and impaired cardiac output (Mielcarek et al., 2014). ANP is also associated with congestive cardiac failure (Mielcarek et al., 2014), which is pertinent for many articles referenced below. These findings suggest mechanisms for how ATTR fibrils may affect cardiac output and electrophysiology.

*The effects of pre-amyloid oligomers on the heart.* Pre-amyloid oligomers originate from proteins such as A$\beta$42 and can induce mitochondrial injury and oxidative stress (Sidorova et al., 2015). Sidorova et al. (2015) showed that rapid pacing of atrial myocytes increased amyloidosis *in vitro* (Sidorova et al., 2015). ANP, a hormone which contributes to development of isolated atrial amyloidosis, was released following rapid activation causing further pre-amyloid oligomer accumulation (Sidorova et al., 2015). Rapid atrial activation contributes to the remodelling observed in atrial fibrillation (AF); therefore, these findings suggest a role for cardiac amyloidosis in cardiac arrhythmogenesis. Rainer et al. (2018) reproduced these findings in mice, observing a 13% increase in florbetapir (a tracer for amyloid aggregation) uptake on PET-CT in the heart due to the presence of amyloid fibrils and phosphorylated desmin (Rainer et al., 2018). From a clinical perspective, desmin cleavage was upregulated twofold in patients with dilated cardiomyopathy, and fourfold in ischaemic cardiomyopathy (Rainer et al., 2018). In a murine study, Sanbe et al. (2005) showed that overexpression of the Desmin-R120G mutation, a mutation responsible for desmin-related cardiomyopathy, results in cardiac amyloidosis and abnormalities on echocardiography (Sanbe et al., 2005). Collectively, the preclinical and clinical evidence suggests that pre-amyloid oligomer formation contributes to the development of arrhythmias, cardiomyopathy and heart failure.

**Amyloid $\beta$ and the cardiovascular system.** A plethora of articles were found which elucidate the direct effects of A$\beta$ on cardiac function. These covered four themes detailing the impact of A$\beta$ on: cardiomyocyte physiology, nitric oxide and endothelial physiology, heart failure and arrhythmogenesis.

*Amyloid $\beta$ peptides negatively affect cardiomyocyte physiology.* Histological evidence localising A$\beta$ in cardiac structures is suggestive of an effect on their function. Numerous studies have demonstrated the presence of APP peptides in human hearts (Arai et al., 1991; Troncone et al., 2016) and rat hearts (Ohgami et al., 1993). Although not all studies on autopsy specimens show the presence of cardiac amyloid plaques (Skodras et al., 1993). Despite this, further histological evidence of cardiac amyloidosis was provided by Hart et al. (2001). This group visualised a heparan sulphate proteoglycan called perlecan, which colocalises with A$\beta$ plaques and NFTs in AD in the hearts of transgenic mice (Hart et al., 2001), suggesting that perlecan and A$\beta$ may be involved in cardiac amyloidosis. Additionally, direct injection of A$\beta$42 into rat hippocampi in another study revealed that cerebral A$\beta$42 may affect systemic organs and predispose to a metabolic syndrome and changes to blood vessels (Kheirbakhsh et al., 2018).

The effect of A$\beta$ on cardiomyocyte Ca$^{2+}$ physiology was explored by later research. Jang et al. (2022) demonstrated the toxicity of A$\beta$40 and A$\beta$42 on endothelial cells, cardiomyocytes and cardiomyocyte mitochondria, with A$\beta$42 inducing more mitochondrial fragmentation than A$\beta$40 (Jang et al., 2022). The detrimental effect of A$\beta$42 on Ca$^{2+}$ concentrations and mitochondrial membranes was supported by another study in rat neurons (Ferreira et al., 2015). Nevertheless, Turdi & colleagues (2009) showed that cardiomyocytes from APPswe/PS1de9 mice, a double transgenic model for AD that overexpresses Swedish mutation APP and PS1, displayed reduced peak shortening amplitudes and maximal velocity of shortening/lengthening *versus* wild-type cardiomyocytes (Turdi et al., 2009). This was discovered alongside lower amplitude Ca$^{2+}$ transients (Turdi et al., 2009), providing further evidence for defective Ca$^{2+}$ cardiomyocyte metabolism in AD.

Aside from Ca$^{2+}$ homeostasis, Jang et al. (2023) showed the negative impact of A$\beta$42 on cardiomyocyte and endothelial cell metabolomics (Jang et al., 2023), and the reduction in mitochondrial respiration due to the presence of APP, A$\beta$40 and A$\beta$42 proteins was evidenced by multiple articles (Jang et al., 2023; Sakamuri et al., 2022; Schmidt et al., 2008). Similarly, numerous studies linked the metabolic effects of A$\beta$42 to contractile dysfunction (Hall et al., 2024; Murphy et al., 2022) and upregulation of ANP (Hall et al., 2024). Interestingly, work by Tang et al. (2023) showed that ANP can interact with A$\beta$ peptides through a process called cross-seeding to halt A$\beta$ plaque formation, where higher concentrations of ANP inhibited amyloidosis to a greater extent. Furthermore, ANP was found to prevent A$\beta$-induced SH-SY5Y (neuroblastoma cell line) cell death. These protective effects translated into worm models, where ANP was cytoprotective and reduced oxidative stress (Tang et al., 2023). These findings suggest that the rise in ANP seen during CVDs may initially protect against AD progression. Perhaps, this protective mechanism becomes overwhelmed as the CVD becomes more severe and a more extreme elevation in ANP may result in the diastolic dysfunction and cardiac remodelling discussed above.

Zhang et al. (2014) showed that a different A$\beta$ fragment, A$\beta$25-35, was also toxic to cardiomyocytes (Zhang et al., 2014). Three potential mechanisms for this were proposed: endoplasmic reticulum stress; phosphorylated ROCK protein disturbing intracellular cytoskeletal scaffolding; and p38 and ERK1/2 MAPK dysregulation, which are involved in apoptosis, proliferation and cytoskeleton organisation (Zhang et al., 2014). The involvement of the endoplasmic reticulum was evidenced by Botteri & colleagues (2018) who showed that BACE1 protein and its breakdown product, sAPP$\beta$, induce endoplasmic reticulum stress and interrupt mitochondrial oxidative phosphorylation (Botteri et al., 2018). Similarly,

Song et al. (2008) showed how A$\beta$42 negatively affects endoplasmic reticulum function in neurons (Song et al., 2008). Conversely, Turdi et al. (2009) showed that protein markers of endoplasmic reticulum stress were similar between cardiomyocytes from APPswe/PS1de9 and wild-type mice (Turdi et al., 2009).

The physiological sequelae of A$\beta$ amyloidosis in the heart were demonstrated by Anwar & Mabrouk (2023), who also related AD amyloidosis to troponin release, an indicator of myocardial damage (Anwar & Mabrouk, 2023). Furthermore, Haase et al. (2013) showed how A$\beta$40, A$\beta$42 and A$\beta$25-35 can affect heart rate and vasoconstriction through acting on adrenergic hormone receptors, suggesting a role of A$\beta$ in tachyarrhythmias and CHD (Haase et al., 2013). However, Krämer et al. (2018) located aggregates of two APP fragments outside the cardiac conduction system, which may suggest a limited electrophysiological effect of A$\beta$, but these aggregates did not associate with cerebral A$\beta$ or NFT deposition (Krämer et al., 2018), meaning that an effect of AD on arrhythmogenesis cannot be ruled out. Independent of its electrophysiological effects, cardiac A$\beta$40 deposition was shown to induce cardiomyocyte apoptosis, reduce contractile function of the heart and increase cardiac remodelling by Elia & colleagues (2023), but at the time of this review this paper has not been peer-reviewed (Elia et al., 2023).

Considering all of this evidence, the detrimental effects of A$\beta$ on the heart are clear, but perhaps only some A$\beta$ fragments connect cardiac dysfunction with AD pathology.

*Amyloid $\beta$ alters nitric oxide physiology and endothelial function.* Another cell type affected by the A$\beta$ accumulation seen in AD is endothelial cells. Two studies revealed how endothelial dysfunction may predispose to cerebral A$\beta$42 deposition (Austin & Katusic, 2020; Cifuentes et al., 2017) and tau phosphorylation (Austin & Katusic, 2016), and A$\beta$ was shown by Oliveira et al. (2011) to inhibit nitric oxide production in retinal neurons (Oliveira et al., 2011). However, for this review, interest was focused on the impact of A$\beta$ peptides on endothelial cell nitric oxide and the heart.

As mentioned earlier, A$\beta$ peptides are toxic to endothelial cells (Jang et al., 2022), and their impact on endothelial cell physiology is confirmed by many studies (Carelli-Alinovi et al., 2016; Chisari et al., 2010; Lamoke et al., 2015; Niwa et al., 2001; Parodi-Rullan et al., 2020; Price et al., 2001; Suo et al., 1997; Sutton et al., 1997). There has been much research investigating the mechanisms underlying A$\beta$-mediated endothelial cell dysfunction. A study by Soriano & colleagues (2003) showed A$\beta$42 and A$\beta$25-35 induce a reduction in endothelial cell metabolic activity which may involve lysosomes (Soriano et al., 2003). A different mechanism for endothelial dysfunction

was shown by Singh Angom et al. (2019); Aβ42 peptide caused endothelial cell senescence via upregulation of vascular endothelial growth factor receptor 1 expression (Singh Angom et al., 2019). Other research indicates that senescent endothelial cells have increased amyloidogenic potential, and a role of AngII in Aβ40 production by these cells (Sun et al., 2018), perhaps strengthening the link between endothelial dysfunction and AD. The role of Aβ-mediated vascular inflammation through interleukin-17 in impairment to vasodilatation was also explored by Vellecco & colleagues (2023) (Vellecco et al., 2023). Many other studies proposed a role for oxidative stress in impairing endothelial cell-mediated vasodilatation (Khalil et al., 2002; Lamoke et al., 2015; Niwa et al., 2001; Park et al., 2005; Suo et al., 1997; Thomas et al., 1997). For example, Park et al. (2005) showed that Aβ40-induced oxidative stress by NADPH oxidases diminishes nitric oxide-mediated vasodilatation (Park et al., 2005). Two other studies agree, showing that Aβ40 induces vasoconstriction through endothelial cell oxidative stress (Khalil et al., 2002; Thomas et al., 1997). Although not using endothelial cells, another study elucidated how the antioxidant resveratrol reduced apoptosis and mitochondrial dysfunction induced by Aβ25-35 (Jang & Surh, 2003). Contradictorily, Stepanichev et al. (2008) showed that Aβ25-35 increased cerebral NOS activity (Stepanichev et al., 2008), which could imply an increase in vasodilatation, or result in cerebral toxicity due to accumulation of nitric oxide. Alternatively, there is evidence for the role of endothelin-1 in AD, due to Aβ40 and Aβ42 (Khalil et al., 2002; Palmer et al., 2012; Palmer et al., 2013). Indeed, administration of the endothelin A receptor antagonist Zibotentan in a different study prevented the Aβ40-induced rise in systolic and diastolic blood pressure in rats (Palmer et al., 2020). This study implies that Aβ40 may affect endothelial function and blood pressure through upregulation of endothelin-1, and highlights a potential use of Zibotentan in managing the cardiovascular complications of AD. Expanding upon the role of endothelin-1 in AD, over-expression of endothelin-1 in mice by Hung & colleagues (2015) has been shown to impair cognition and lead to brain changes characteristic of AD (Hung et al., 2015).

The effect of Aβ in increasing oxidative stress and reducing cellular metabolism may suggest an impact on mitochondrial function (Suo et al., 1997). So far, Aβ-induced endothelial cell toxicity was shown to occur through multiple mechanisms and impair nitric oxide-mediated vasodilatory responses. Stemming from these findings, several studies implicate Aβ-induced nitric oxide dysfunction in mitochondrial failure. Keil et al. (2004) showed that Aβ42 increased nitric oxide levels *in vitro*, and cells expressing APP genes, particularly the Swedish mutant, showed higher NOS activity, depolarisation of the mitochondrial membrane potential and inhibition of the mitochondrial respiratory complexes (Keil, Bonert, Marques, Scherping et al., 2004; Keil, Bonert, Marques, Strosznajder et al., 2004), which was nullified after inhibition of γ-secretase (Keil, Bonert, Marques, Scherping et al., 2004). Another study also illustrated the role of γ-secretase in cardiac mitochondria, and suggested how dysfunction of hypoxia inducible factor-1 gene may predispose to mitochondria-mediated cell death in AD (Hayashi et al., 2012). Furthermore, Solesio et al. (2018) highlighted a role for carbonic anhydrase in mediating the mitochondrial membrane depolarisation and cell death due to Aβ40 and Aβ42 (Solesio et al., 2018). Perhaps, Aβ-induced mitochondrial toxicity may involve γ-secretase and carbonic anhydrase.

Murine models also provide evidence for an alternative pathway for endothelial dysfunction in AD involving ion movement and the subsequent effects on cognition. Firstly, it was shown that an increased carotid artery systolic blood pressure associated with poorer learning and memory abilities, reduced endothelial function, and endothelial cell apoptosis (De Montgolfier et al., 2019). The direct effects of Aβ40 on arterial vasoconstriction were demonstrated by Suo et al. (2000), who infused this peptide into hypotensive rats, although this effect was not replicated in normotensive or hypertensive animals (Suo et al., 2000). Taylor et al. (2022) visualised CAA in APP-overexpressing mice and demonstrated defective endothelial-mediated vasodilatation through reduced inward rectifier (Kir2.1) $K^+$ current density in cerebral arteries (Taylor et al., 2022). Similarly, the impaired $Ca^{2+}$-mediated vasodilatation seen during CAA was explored by Peters et al. (2022). It was found that incubation of murine cerebral arteries with Aβ40 reduced $Ca^{2+}$ entry through NMDA receptors in endothelial cells, and the frequency of $Ca^{2+}$ transients inside the cells declined. This translated to reduced vasodilatation in posterior communicating arteries from wild-type C57bl6 mice. Moreover, an AD mouse model was used to illustrate perivascular Aβ plaques in pial arteries, and these vessels were less sensitive to NMDA-induced vaso-dilatation (Peters et al., 2022). Another murine study by Meakin & colleagues (2020) confirmed that levels of Aβ42 and BACE1 proteins were increased in the aortas of obese mice, which impaired phosphorylation of eNOS, inhibiting vasodilatation (Meakin et al., 2020). Hyper-tension present in mice fed high-fat diets was reversed through inhibition of BACE1. The authors also performed an observational study in patients with diabetes and discovered that these patients demonstrated endothelial dysfunction alongside increased plasma Aβ42 (Meakin et al., 2020). Similarly, patients with AD were found to have impaired vasodilatation compared with patients without dementia, by Khalil et al. (2007), with greater impairment seen with more severe cognitive dysfunction (Khalil et al., 2007). These publications illustrate how Aβ

peptides can influence $Ca^{2+}$ signalling and $K^+$ channels in endothelial cells and hence blood pressure regulation and cognition.

In the context of CVD, Kitazume et al. (2012) showed that endothelial cell and platelet phenotypes, similar to those seen in myocardial infarction, released APP770, an APP isoform distinct from sAPP$\alpha$ and sAPP$\beta$ (Kitazume et al., 2012). Indeed, patients with myocardial infarction demonstrated higher circulating APP770, and the authors showed that concentrations of sAPP$\alpha$ are increased in a rat model of myocardial infarction. While these findings suggest a role of APP processing in CVD, APP770 did not correlate to AD pathology in patients (Kitazume et al., 2012). This is similar to a finding by Alifier et al. (2020), who showed that tau did not associate with myocardial infarction (Alifier et al., 2020). Perhaps, APP processing in endothelial cells contributes to cardiovascular dysfunction independent of cerebral amyloidosis.

Overall, there appears to be an association between A$\beta$, dysfunctional NO-mediated vasodilatation and subsequent CVD both clinically and in animal studies. The release of amyloidogenic peptides from cells involved in myocardial infarction demonstrates how A$\beta$, and by proxy AD pathology, can predispose to CVDs.

*Amyloid $\beta$ peptides promote atherosclerosis and heart failure.* A single study was found that aimed to explore the role of AD in atherosclerotic pathophysiology, specifically the greater risk of death associated with A$\beta$40 in patients with CVD (Stamatelopoulos et al., 2015). Additionally, A$\beta$40 associated with increased pulse wave velocity, burden of atherosclerosis in peripheral arteries and CHD confirmed on coronary angiography (Stamatelopoulos et al., 2015). Conversely, the study by Meakin et al. (2020) showed that major arteries of BACE1-knockout and wild-type mice infused with A$\beta$42 showed no histological evidence of atherosclerosis (Meakin et al., 2020). This suggests that different A$\beta$ peptides alter cardiovascular pathology differently, or responses to A$\beta$ are different between animals and humans.

Aside from atherosclerosis, other studies describe how A$\beta$ can predispose to heart failure phenotypes (Aishwarya et al., 2024; Greco et al., 2017; Sanna et al., 2019; Yousefirad et al., 2016b; Zhu et al., 2023). Yousefirad et al. (2016b) described the negative inotropic effect of the fragment A$\beta$22-35, independent of changes in heart rate and coronary blood flow (Yousefirad et al., 2016b). The same research group reproduced these effects for A$\beta$42 peptides, but A$\beta$42 had extra effects on repolarisation, heart rate and coronary blood flow (Yousefirad et al., 2016a). Given findings mentioned earlier (Haase et al., 2013), one may assume that different fragments of A$\beta$ affect cardiac rate, rhythm and contractility in different ways. Additionally, Aishwarya et al.

(2024) performed echocardiography on APPswe/PS1de9 mice and wild-type mice (Aishwarya et al., 2024) and measured parameters including ejection fraction, the volumetric fraction of blood exiting the left ventricle during systole, and fractional shortening, which is the change in left ventricular dimensions between the end of diastole and the end of systole (Murphy et al., 2022). While these two parameters were similar between groups, APPswe/PS1de9 mice demonstrated filling defects and ventricular dyssynchrony alongside extracellular matrix remodelling and mitochondrial dysfunction (Aishwarya et al., 2024).

These preclinical findings are corroborated by studies in human participants. Notably, the landmark study by Troncone et al. (2016) showed worse diastolic function in patients <65 years old with AD and increased thickness of the left ventricle in patients >80 years old, a feature observed in classic cardiac amyloidosis (Troncone et al., 2016). Although these findings are only observed in specific age categories, this still provides evidence that AD affects the heart in numerous ways as an individual ages. In four cardiac specimens from AD patients, the presence of A$\beta$40 and A$\beta$42 were confirmed (Troncone et al., 2016), providing evidence that AD affects cardiac function through direct cardiac amyloidosis. Vetrano et al. (2016) corroborated these findings in patients with Down's syndrome, which is a risk factor for early-onset AD (Vetrano et al., 2016). Additionally, Jin & colleagues (2017) showed that patients diagnosed with AD demonstrated lower ejection fraction on echocardiography irrespective of the patient's age (Jin et al., 2017). Similar findings were observed in the Rotterdam study, but the effect of A$\beta$40 on LV was only statistically significant in men, implying that the effect of A$\beta$40 on LVEF and LV mass is more severe in males than females (Zhu et al., 2023). Alternatively, Koemans et al. (2024) showed that female patients with cognitive impairment experienced more severe A$\beta$ deposition than male patients (Koemans et al., 2024), but this study did not assess the impact on cardiovascular function. Furthermore, Sanna et al. (2019) showed that AD patients exhibited increased diastolic dysfunction and increased wall thickness, and there were no other echocardiographic abnormalities or arrhythmia on electrocardiography (Sanna et al., 2019).

When trying to determine the pathophysiology of these changes, many studies outline a key role for the BACE1 protein. Older research shows similar levels of APP, BACE and presenilin-1 messenger RNA in the hearts of patients with and without AD (Yasojima et al., 2001), and lower concentrations of BACE protein in heart specimens than brain specimens from murine models of AD (Rossner et al., 2001). Nonetheless, the role of BACE in cardiovascular pathology was explored by Greco et al. (2017), who demonstrated upregulation of sense and antisense BACE1 RNA, two long non-coding RNAs that are trans-

cribed from opposite DNA strands of the BACE1 gene, in patients with heart failure and in mouse models of myocardial infarction (Greco et al., 2017). Antisense BACE1 has been shown to increase the expression of sense BACE1, and the authors showed *in vitro* that anti-sense BACE1 increases the concentration of A$\beta$40 in cardiomyocytes and endothelial cells resulting in cell death (Greco et al., 2017). Interestingly, an alternative role for BACE1 was displayed in a murine study looking at the relationship between neuronal BACE1 protein levels and diabetes mellitus (Plucińska et al., 2016). A diabetic phenotype and defective cerebral glucose metabolism were noted after BACE1 knock-in. Indeed, BACE1 knockout is known to be protective against diabetes and obesity in mice. Although cardiac glucose metabolism was unaffected, this study suggests that increased BACE1 expression, such as during AD, can pre-dispose to CVD indirectly through creating a diabetic phenotype. Additionally, BACE1 knock-in induced end-oplasmic reticulum stress within the hypothalamus and altered its endocrine function (Plucińska et al., 2016), which could suggest that BACE1 may predispose to an AD endocrinopathy, as discussed later.

In summary, A$\beta$40 and A$\beta$42 appear to affect cardiac morphology and function, and the BACE1 protein may mediate this association. Also, studies in mice and humans affected by AD pathology show cardiac amyloidosis and subsequent diastolic dysfunction.

*Amyloid $\beta$ amyloidosis has an effect on cardiac arrhythmias.* There is evidence to suggest that AF can increase the risk of cerebral A$\beta$ and tau amyloidosis as well as neuroinflammation (Du et al., 2022) and, vice versa, AD has been shown to impact cardiac electro-physiology. For example, Sanna et al. (2019) reported that low-voltage QRS complexes, an electrophysiological marker of cardiac amyloidosis, were more common in patients with AD compared to those without (Sanna et al., 2019). This suggests that AD pathology may affect cardiac electrophysiology, a finding which is reproduced by two papers from the ARIC-PET amyloid imaging study (Johansen et al., 2020, 2022). Atrial cardio-pathy as a composite measure correlated with increased imaging evidence of cerebral A$\beta$ deposition. The only component of atrial cardiopathy to associate with A$\beta$ plaque deposition was left atrial volume index (Johansen et al., 2020). In addition, premature atrial contractions slightly associated with increased A$\beta$ amyloidosis, but this was not true for AF or atrial tachycardia (Johansen et al., 2022). These cross-sectional studies in human patients suggest a limited effect of A$\beta$ on arrhythmogenesis in patients.

However, animal and *in vitro* studies imply a greater effect of A$\beta$ on arrhythmogenesis. Turdi et al. (2009) discovered that APPswe/PS1de9 mice had higher resting

heart rates, shorter R–R intervals, and reduced R wave and QRS complex amplitudes than their wild-type counter-parts (Turdi et al., 2009), suggesting potential influences of cerebral amyloidosis on heart rhythm. Moreover, a fundamental science study outlined the effect of AD pathology on cardiac ion channels (Agsten et al., 2015). KCNQ1 (also known as K$_v$7.1) are K$^+$ channel pore-forming $\alpha$-subunits that are responsible for the rapid component of the delayed rectifier current (I$_{Ks}$) involved in cardiac repolarisation, dysfunction in which is a cause of long QT syndrome. Agsten et al. (2015) showed that the BACE1 protein increased response to depolarisation and reduced I$_{Ks}$ in cardiomyocytes, suggesting a potential mechanism of BACE1-induced arrhythmia, but this effect was adenosine triphosphate (ATP)-dependent (Agsten et al., 2015). Although not specific to KCNQ1 K$^+$ channels, one *in vitro* study demonstrated that nicorandil, which opens ATP-dependent K$^+$ channels, protects against apoptosis in AD (Kong et al., 2013). Furthermore, Sachse et al. (2013) demonstrated that KCNE1, a protein that associates with KCNQ1 to form I$_{Ks}$, is cleaved by the BACE1 protein, and reduces cardiac repolarisation *in vitro* (Sachse et al., 2013). These studies highlight a role for K$^+$ channels in AD, and further research is required to understand whether pharmacologically or genetically modulating these channels may be therapeutic in AD.

However, clinical trials suggest a negligible effect of AD on heart rhythm. A small randomised study in elderly patients investigated JNJ-54861911, a BACE1 inhibitor aiming to reduce cerebral A$\beta$ production, and demonstrated that this agent prolongs the QT interval in patients at doses six times higher than the therapeutic dose of 25 mg/day (Timmers et al., 2018). These findings imply that blocking the activity of the BACE1 protein may predispose to, rather than inhibit, arrhythmogenesis and suggests that this drug is a safe option for managing AD at therapeutic doses. Furthermore, an analysis of three phase I/II studies by Vormfelde et al. (2020) showed that umibecestat, another BACE1 inhibitor, had no effect on the QT interval and did not affect other parameters on electrocardiography (Vormfelde et al., 2020). While this study also shows that BACE1 inhibition may be safe for use in AD, it suggests little effect of the BACE1 protein on cardiac electrophysiology. Despite this, these studies included healthy participants; the MIND-China study showed that corrected QT and JT intervals on electro-cardiography associated with clinical AD and plasma A$\beta$40 levels, suggesting an association with impaired cardiac repolarisation and AD (Mao et al., 2023). It is worth noting that AD patients in this study were more likely to be female and have CHD, which may confound this relationship.

Overall, preclinical studies suggest that amyloidogenic processing may predispose to arrhythmogenesis, particularly long QT syndrome, but the clinical evidence

conflicts regarding this conclusion and suggests a more limited association.

**Tau protein and the cardiovascular system.** Significantly fewer articles were retrieved detailing the impact of tau protein on cardiovascular pathology. Two of these were referenced above: Moore et al. (2022) described no association between p-tau and LV mass index (Moore et al., 2022) whereas Hamasaki et al. (2022) detected positive associations between cardiac ATTR and cerebral NFT burden (Hamasaki et al., 2022). One murine study by Luciani et al. (2023) showed that cardiac tauopathy resulted in diastolic impairment, left atrial enlargement and increased left atrial pressure (Luciani et al., 2023). Additionally, echocardiography in patients showed that p-tau was nearly twice as prevalent in hearts from patients with dilated cardiomyopathy compared to patients without this condition (Luciani et al., 2023). Hearts from AD patients exhibited higher t-tau and had threefold more prevalent p-tau than hearts from patients without AD. These authors also showed that tauopathy alters the intracellular cytoskeleton *in vitro* and suggested that this may affect protein clearance inside the cell, thus predisposing to further amyloidosis (Luciani et al., 2023). Overall, these findings suggest an involvement of p-tau in the development of cardiomyopathies in patients.

**Alzheimer's disease genes and cardiac physiology.** Dave et al. (2023) evidenced the presence of an altered cardiac phenotype in a transgenic AD mouse model. Specifically, transgenic mice with APPswe/PS1de9 mutations showed increased heart weight compared with wild-type mice, which may indicate cardiac hypertrophy, and higher expression of genes associated with cardiac pathology (Dave, Judd et al., 2023). In another study, these mutations were shown to affect cardiac chamber diameters, systolic and diastolic function, and mechanical properties of cardiomyocytes which indicate disturbed $Ca^{2+}$ metabolism (Zhu, Liu et al., 2022). Interestingly, insertion of a transgene for mitochondrial aldehyde dehydrogenase nullified these effects, implying that APPswe/PS1de9 genes may be cardiotoxic through impairing mitochondrial enzyme activity. The effects of APPswe/PS1de9 mutations on heart metabolomics were explored by Zheng et al. (2019), who showed no difference in amino acid levels between APPswe/PS1de9 and wild-type mice for up to 10 months of age (Zheng et al., 2019).

Bioinformatics analysis has shown similar genes to be associated with cardiovascular pathology as well as AD (Murakami & Lacayo, 2022). Similarly, genetic analysis of human participants by Wen & colleagues (2022) showed that genes associated with widespread cortical atrophy were expressed in the heart as well as in the brain (Wen et al., 2022). This suggests that

genes associated with AD pathology may also affect cardiac function. A transcriptomic analysis by Zhang et al. (2024) of brains from AD patients identified nine potential genes which may predispose to AD (Zhang et al., 2024). The *SHC2* gene (encoding SHC Adaptor Protein 2), implicated in sympathetic neuron degeneration, was identified suggesting a genetic influence on autonomic nervous system dysfunction in AD. In addition, *GNB5* was identified and, when co-expressed with APPswe/PS1de9 in transgenic mice, substantially potentiated cerebral A$\beta$ and NFT deposition. Loss of function mutations in *GNB5* are known to predispose to a syndrome of neurodevelopmental and language delay alongside cardiac arrhythmias (Zhang et al., 2024). A plausible explanation may be that individuals with this syndrome are more likely to develop AD and that genes predisposing to AD also predispose to arrhythmias, although, this was not explored by this paper.

The final paper regarding AD genetics is a genotypic analysis of the ARIC study, specifically of the APOE-$\varepsilon$4 allele (Selvaraj et al., 2022). After accounting for confounders, there were no differences in heart failure prevalence or LVEF, diastolic function or chamber morphologies, and A$\beta$ did not predispose to the development of heart failure. This study suggests no cardiovascular effect of a common gene involved in AD pathophysiology (Selvaraj et al., 2022). Knockout of the APOE gene is a known model of atherosclerosis in mice (Lewandowski et al., 2020). Substantial evidence supports a role for APOE in modulating the aggregation and clearance of A$\beta$, neurodegeneration, tau pathology and inflammation. Hence, APOE knockout mice would be a good model to study the interaction of amyloidosis with cardiovascular risk factors in exacerbating cardiac complications of AD. However, our systematic review did not capture any such studies.

In summary, AD genes predispose to cardiovascular sequelae in animal models, and many AD-associated genes were shown to be associated with the development of CVD in humans.

**Autonomic dysregulation in Alzheimer's disease.** Research articles describing autonomic dysregulation in AD were categorised as those describing AD pathology affecting the brainstem nuclei, and those describing the subsequent effects on the SNS and PNS.

*Alzheimer's amyloidosis in brainstem autonomic nuclei.* A$\beta$ and tau amyloidosis has been detected in many parasympathetic and sympathetic nuclei (Parvizi et al., 2001; Tian et al., 2022), as well as the periaqueductal grey matter, known to regulate blood pressure and heart rate (Parvizi et al., 2000). Interestingly, Tian et al. (2022) reported an absence of A$\beta$42 staining within the medulla oblongata in patients with AD (Tian et al.,

2022), and another study showed mostly NFTs in the medullary reticular formation and the pontine nuclei which communicate with this region (Rüb et al., 2001). This predominance of NFT amyloidosis was also found in the spinal cords of AD patients (Guo et al., 2016), which may contribute to abnormal outputs from the PNS or SNS. Indeed, the aforementioned study by Arai et al. (1991) localised A$\beta$ plaques in peripheral nerves (Arai et al., 1991).

Linking this to cardiovascular complications of AD, Jacobs et al. (2022) showed that lower volume of each section of the brainstem associated with increased cerebral A$\beta$ amyloidosis (Jacobs et al., 2022). Although, there were no associations between brainstem volumes and tau amyloidosis, and AD-related amyloidosis did not associate with hippocampal volume (Jacobs et al., 2022), perhaps implying that the amyloidosis observed in the brainstem is not related to a typical AD phenotype seen in the majority of patients.

*Sympathetic and parasympathetic nervous dysfunction in Alzheimer's disease.* Elia et al. (2023) demonstrated a reduction in cerebral and cardiac brain-derived neurotrophic factor (BDNF) in Tg2576 mice, a model of cerebral amyloidosis (Elia et al., 2023), which predisposed to a reduction in cardiac sympathetic nerve fibres. Incubation of cardiomyocytes with A$\beta$ reduced levels of BDNF and growth-associated protein 43, a marker of neuronal regeneration (Elia et al., 2023). The reduction in sympathetic nerve fibres and neuronal regeneration suggest reduced SNS activity in AD. Another murine study by Lai & colleagues (2019) strengthens the plausible association between SNS activity in AD and arrhythmia formation (Lai et al., 2019). These authors examined the effects of A$\beta$40 on sympathetic preganglionic neurons, and discovered how this peptide exacerbates depolarisation in these neurons through potentiating the action of NMDA on its receptors. These receptors in these neurons are vital for modulating central control of sympathetic function (Lai et al., 2019), so their augmentation by soluble A$\beta$ oligomers may explain the involvement of A$\beta$ in modulating cardiovascular function.

Furthermore, Chen et al. (2023) discovered that in transgenic AD mice, R-R intervals, the total power of heart rate variability (HRV) and high-frequency power of HRV were reduced during sleep and wakefulness at different disease stages (Chen, Kwok et al., 2023). These parameters correspond to lower cardiac autonomic and vagal activity, indicating impaired PNS activity in AD (Chen, Kwok et al., 2023). A third study demonstrated that components of $\gamma$-secretase are required for dendritic growth of sympathetic neurons in the rat heart (Karunungan et al., 2023). This suggests that $\gamma$-secretase may be involved in growth of sympathetic neurons, and thus

predispose to sympathetic hyperactivity in the heart and arrhythmogenesis, which contradicts the sympathetic downregulation shown by Elia et al. (2023). The final preclinical study was performed by Tayler et al. (2018) who showed that A$\beta$ infusion potentiated pre-existing hypertension in rats but had no effect in normotensive rats (Tayler et al., 2018). While HRV was not affected by A$\beta$40 infusion, a reduced sensitivity of the baroreflex response to blood pressure changes, measured by baroreflex gain, was noted (Tayler et al., 2018) implying that AD may precipitate autonomic dysregulation. These preclinical studies suggest suppressed PNS activity, and different degrees of SNS dysregulation, which would predispose to arrhythmogenesis and blood pressure dysregulation in AD.

Stemming from this, a clinical study in patients with cognitive impairment due to a vascular aetiology found that biomarkers of neuronal injury and AD were not associated with blood pressure variability (Starmans et al., 2024). Interestingly, it is known that variability in blood pressure predisposes to dementia (Starmans et al., 2024), and the findings of this paper suggest no effect of dementia on blood pressure changes, contradicting the findings of Tayler et al. (2018). Furthermore, Miller et al. (2008) investigated baroreflex dysregulation in patients with carotid sinus hypersensitivity, whereby blood pressure and heart rate fall (Miller et al., 2008). These patients demonstrated tauopathy in the brainstem autonomic nuclei involved in cardiovascular reflex processing (nucleus tractus solitarius, nucleus ambiguus, dorsal motor nucleus of the vagus), but diagnosis of AD in these patients did not correlate to amyloidosis in autonomic nuclei (Miller et al., 2008). Despite its small size, this study suggests amyloidosis-induced autonomic dysregulation is independent of AD diagnosis. Similarly, Xue et al. (2023) showed that patients with AD risk factors, specifically obstructive sleep apnoea, but who are not diagnosed with AD, can experience autonomic dysfunction (Xue et al., 2023). Lower HRV correlated with cognitive impairment and more cerebral t-tau and p-tau deposition (Xue et al., 2023); findings supported by another study in patients with mild cognitive impairment (Díaz-Román et al., 2021).

Although not performed in patients with dementia or AD, Lohman et al. (2024) demonstrated that connectivity in the central autonomic network, which consists of the cortical and subcortical regions controlling respiratory, cardiovascular and other autonomic systems, positively correlated with plasma A$\beta$42/A$\beta$40 ratio (Lohman et al., 2024). Accordingly, dysfunction to the ANS may relate to biomarker changes indicative of AD. Consistently, Molloy et al. (2023) showed that higher resting heart rate, perhaps implying SNS overactivity or PNS underactivity, correlates with worse cognitive performance. Specifically, a lower HRV was seen in patients with

abnormal amyloid/tau ratio who exhibit poorer accuracy on cognitive testing (Molloy et al., 2023). A different study compared cardiac electrophysiology during task switching in patients with pathological amyloid/tau ratios with those with normal ratios (Arechavala et al., 2021). It was shown that those with abnormal ratios had a drop in low frequency domain of HRV, and a drop in the standard deviation between the R–R intervals, between resting and task states. This implies an imbalance between the SNS and PNS in patients with abnormal amyloid/tau ratios. The R–R intervals dropped during the task in both groups but to a greater degree in the group with a pathological amyloid/tau ratio (Arechavala et al., 2021). This implies that HRV could be used to identify patients with some degree of cognitive impairment, underlying AD pathology, and a malfunctional brain–heart axis, but more research is needed to demonstrate the feasibility of this for clinical practice.

In adults with preclinical AD, Santos et al. (2017) performed electrocardiographic measurements of respiratory sinus arrhythmia, considered an index of vagal tone, and the ratio between HRV and respiratory sinus arrhythmia during a cognitive stressor (Santos et al., 2017). Patients with imaging evidence of cerebral A$\beta$ showed no changes in vagal ratio or respiratory sinus arrhythmia during cognitive stress, in contrast to individuals without A$\beta$ deposits (Santos et al., 2017), suggesting impairment to the PNS in preclinical AD. This association is bidirectional, as shown by Min et al. (2023). Participants utilising slow-paced breathing techniques to increase HRV exhibited decreased A$\beta$42 and A$\beta$40 levels; the opposite effect was seen in those trying to minimise their heart rate oscillations. The effect on tauopathy appears to be age-dependent, with increases in t-tau due to HRV reduction in younger adults only, but HRV reduction increased p-tau in older adults only (Min et al., 2023). Mechanisms suggested by the authors regarding how slow-paced breathing reduced A$\beta$ production, increased renal excretion of A$\beta$, or increasing clearance of cerebral A$\beta$ and tau fibrils involved cardio-respiratory feedback mechanisms stimulating vagal activity and reducing adrenergic stress (Min et al., 2023). The same research group are conducting a second study to reproduce these findings, and also determine whether slow-paced breathing affects hippocampal volumes and cognition (Nashiro et al., 2024).

Overall, Min et al. (2023) revealed a complex interplay between autonomic dysfunction, AD pathology and age and is in accordance with the previous studies showing that AD predisposes to parasympathetic dysfunction.

**Hormonal alterations in Alzheimer's disease.** The final theme identified from the literature search pertained to the endocrine system in AD. Of note, the HPA axis was shown to be affected by AD pathology.

*HPA axis dysfunction relates to cardiovascular sequelae of Alzheimer's disease.* Analysis of postmortem brains from AD patients revealed reduced neuron populations and intracellular organelle dysfunction in numerus hypothalamic nuclei (Baloyannis et al., 2015). Further evidence for the involvement of the HPA axis comes from Arai et al. (1991), who discovered AD pathology in the pituitary and adrenal glands (Arai et al., 1991). This provides histological evidence of AD endocrinopathy, but no evidence of any cardiovascular sequelae.

The function of the hypothalamus has been shown to be altered in the pathophysiology of AD in murine studies. For instance, disruption to the circadian rhythm controlled by the suprachiasmatic nucleus, a phenomenon observed in patients with AD, was replicated through hippocampal administration of A$\beta$31-35 to C57BL/6 mice (Wang et al., 2016). Moreover, this study showed an alteration to the rhythmic expression of circadian regulator genes in the heart (Wang et al., 2016), which might play a role in AD-induced CVD. Activation of the HPA axis in transgenic AD mice has been shown by Hebda-Bauer et al. (2013). These mice, particularly males, demonstrated more hypothalamic glucocorticoid receptors and reduced levels of corticotropic-releasing hormone. (Hebda-Bauer et al., 2013). Usually, this suggests negative feedback to reduce circulating corticosterone when it is present in excess, but AD mice exhibited similar levels of corticosterone to their wild-type counterparts in this study (Hebda-Bauer et al., 2013). Perhaps, the findings of these studies represent subclinical hormonal dysfunction in early-stage AD pathology. Similarly, Pedersen et al. (1999) concluded that transgenic AD mice were unable to maintain normoglycaemia during periods of stress, representing desensitisation to the effects of corticosterone and thus dysfunction of the HPA axis (Pedersen et al., 1999). Furthermore, Morgese et al. (2014) administered soluble A$\beta$ peptide to rats resulting in poorer passive avoidance test performance, where rats must learn to refrain from entering a previously punished area. This occurred in conjunction with reduced plasma corticosterone, implying HPA dysfunction in symptomatic AD (Morgese et al., 2014). These results suggest that AD may cause an endocrinopathy characterised by adrenal insufficiency.

On the other hand, Touma et al. (2004) discovered higher plasma corticosterone concentrations and increased faecal corticosterone metabolites in APP-expressing mice compared to wild-type mice (Touma et al., 2004), which demonstrates the presence of hypercortisolism in AD. Interestingly, the sympathetic-adrenomedullary system was unaffected by APP gene expression, and cerebral A$\beta$ burden did not correlate with endocrinopathy (Touma et al., 2004). While this could suggest no association between AD and endocrinopathy, it could also be explained by specific

dysfunction of the HPA axis causing increased cortisol secretion. Lv et al. (2020) provided similar results, however, showing A$\beta$42 infusion increased markers of cortisol activity and cognitive impairment (Lv et al., 2020). Furthermore, Hendrickx et al. (2021) related increased activity of corticosterone to elevations in blood pressure, increased arterial stiffness, and aortic non-compliance in transgenic mice over-expressing APP, providing a link between hypercortisolaemia and cardiovascular complications of AD (Hendrickx et al., 2021). Intriguingly, metabolites of adrenaline and noradrenaline in the urine were increased in these mice, suggesting an impairment to adrenergic activity in AD (Hendrickx et al., 2021). Overall, some studies show that the function of steroid hormones is reduced in AD mice, whereas others demonstrate an effect of hypercortisolism on the cardiovascular system (Hendrickx et al., 2021; Lv et al., 2020; Touma et al., 2004). Potentially, AD endocrinopathy exists on a spectrum with cortisol levels depending on the extent of A$\beta$ and tau amyloidosis.

To clarify the relationship between cortisol and CVD, clinical studies yielded by our search were analysed. Popp et al. (2015) showed that patients with AD exhibit higher CSF cortisol than dementia-free controls which correlates to symptom severity, but the presence of AD did not affect plasma cortisol levels (Popp et al., 2015). Likewise, Pietrzak et al. (2017) showed that in patients with imaging evidence of cerebral A$\beta$ deposition, high circulating cortisol predisposed to worse cognitive outcomes than low cortisol levels (Pietrzak et al., 2017). Thirdly, mini mental state examination scores in patients with AD negatively correlated with plasma cortisol levels, which may be due to intracellular signalling molecule disturbances (Vasantharekha et al., 2024). Unfortunately, no clinical studies have shown how AD hypercortisolaemia might predispose to CVD. Perhaps the mechanisms proposed by Hendrickx et al. (2021) in mice are present in humans, but research is needed to explore this further.

*Alzheimer's pathology downregulates the effects of melatonin.* The only other CVD-relevant hormone affected in AD found from the search was melatonin. Cecon et al. (2015) revealed how A$\beta$40 and A$\beta$42 reduce melatonin production by the pineal gland and A$\beta$42 downregulates melatonin receptors and their ligand binding site on endothelial cells (Cecon et al., 2015). This implies a potential effect of melatonin dysregulation on the cardiovascular system. Developing this hypothesis, Wang et al. (2020) showed that blood melatonin concentrations are reduced in AD patients alongside defective ventricular contraction (Wang et al., 2020), implying melatonin dysfunction may lead to cardiac dysfunction. The same study showed similar findings in AD transgenic mice, alongside also impairing cardiomyocyte contraction and Ca$^{2+}$ metabolism. Melatonin

supplementation negated these abnormalities, through increasing mitochondrial aldehyde dehydrogenase activity (Wang et al., 2020) similar to the findings above from Zhu et al. (2022). Finally, melatonin supplementation to transgenic mice over-expressing APP by Feng et al. (2004) reduced A$\beta$ deposition in the frontal cortex, providing more evidence that melatonin may be protective in AD.

Overall, the effects of AD on cortisol and melatonin seems to negatively affect cardiac function and endothelial cell physiology, providing another mechanism by which AD may precipitate CVD.

## Discussion

**The bidirectional relationship between AD and cardiovascular disease.** From all the evidence analysed, it is clear that cerebral hypoperfusion increases cerebral A$\beta$ and tau amyloidosis, neurodegeneration, AD gene expression and cognitive dysfunction. Indeed, assessing one's cardiovascular risk may be useful in predicting the development of future AD pathology and cognitive decline. Some effects appear to be sex specific, where female mice and humans demonstrate more severe AD neuropathology and cognitive impairment. Cardiac amyloidosis of mutated ATTR and pre-amyloid oligomers affect cardiomyocyte physiology, cardiac remodelling and hypertrophy. Preclinical evidence suggests the existence of cardiac electrophysiological remodelling and aortic valve disease in cardiac amyloidosis. On a cellular level, amyloidosis of A$\beta$, and to an extent tau protein, is toxic to cardiomyocytes, endothelial cells, pericytes and autonomic neurons. A$\beta$ upregulates endothelin-1, impairing endothelial function and raising blood pressure (Palmer et al., 2020) – this potential causative role of A$\beta$ and ET1 in hypertension as a cardiovascular complication of AD warrants further investigation. In the brain, A$\beta$ constricts capillaries through the generation of reactive oxygen species, enhanced endothelin-1 release, and contraction of capillary pericytes, leading to a reduction of cerebral blood flow (Nortley et al., 2019). In detail, A$\beta$42 activates NOX4 (NADPH oxidase subtype 4) expressed in endothelial cells and pericytes, the oxidative stress then drives the increased ET1 production by endothelial cells, activating ET1A receptors on pericytes leading to Ca$^{2+}$ dependent constriction and decrease of capillary diameter (in postmortem brain tissue, the capillary diameter in AD brains was reduced by up to 8.1% compared with controls). Similarly, in the heart, elevated A$\beta$ and endothelin-1 levels in AD may drive microvascular constriction, reducing coronary blood flow at the capillary level and contributing to Alzheimer's-related cardiac complications, including maladaptive tissue remodelling and corresponding changes in contractility. In addition,

cardiovascular complications of AD may result from the AD genotype and malfunctional nitric oxide physiology, with detrimental effects on mitochondria and peripheral vasodilatation, and the $\gamma$-secretase enzyme appears to be key in mediating these relationships. These findings were reproduced in clinical studies, supporting the association between AD and CVD in patients. In particular, AD - and specifically the BACE1 protein - may be responsible for impairing cardiac output resulting in a heart failure phenotype. The effects of AD pathology on cardiac $I_{Ks}$ also implies a detrimental impact on cardiac repolarisation and QT interval, but this has only been shown in preclinical studies to date. The role of certain AD genes, especially *SHC2* and *GNB5*, needs to be investigated in future studies as these may underlie further associations between AD, autonomic control of the heart and arrhythmogenesis.

Brainstem and spinal cord $A\beta$ and tau amyloidosis may underlie the autonomic dysregulation observed in AD. Confusingly, research suggests that sympathetic fibre degeneration and dendritogenesis may contribute to arrhythmogenesis, and more research is required to explore the consequences on cardiac arrhythmogenesis. As demonstrated in the Supplementary Table, multiple studies have explored how AD pathology and amyloidosis could increase cardiac fibrosis (Elia et al., 2023; Krämer et al., 2018; Mielcarek et al., 2014; Turdi et al., 2009; Zhu, Liu et al., 2022), which in principle could contribute to a substrate for arrhythmogenesis. Finally, AD endocrinopathy involves dysfunction of cortisol and loss of the protective effects of melatonin. The endocrinopathy in AD may exist on a spectrum ranging from adrenal insufficiency to hypercortisolism. While postmortem analysis and animal studies provide strong evidence for hypothalamic dysfunction, disrupted circadian regulation, and altered glucocorticoid signalling, the implications for CVD remain unclear. Only one study related HPA axis dysfunction to arterial stiffness and poor aortic compliance, and thus more research is needed to explore the role of cortisol in cardiovascular complications of AD. As the majority of cited research was observational, all associations described in this review may be bidirectional, and it cannot be said for certain whether AD pathology causes CVD, or vice versa.

**Comparison with other reviews.** A strength of the current review is its systematic nature. However, other published reviews have explored the brain–heart axis. In agreement with our review, Stakos et al. (2020) explored the $A\beta$ hypothesis in relation to CVD and reported an effect of $A\beta$ on pulse wave velocity and markers of peripheral atherosclerosis, as well as cardiotoxic effects of BACE1 and $A\beta40$ proteins (Stakos et al., 2020). However, they covered many aspects of CVD missed by our review. For instance, the presence of $A\beta$ inside atherosclerotic plaques, how $A\beta$ potentiates atherosclerosis through matrix metalloproteinase release and the telomere shortening and ageing of blood vessels induced by $A\beta40$ (Stakos et al., 2020). Our search criteria may have been too specific, and inclusion of terms such as 'artery' may have identified more papers.

A narrative review by Tini et al. (2020) developed the cerebral hypoperfusion hypothesis by introducing the concept of a cerebral metabolic crisis, resulting in acidosis, oxidative stress, and subsequent tau hyperphosphorylation (Tini et al., 2020). Our review does not explore this hypothesis in as much detail. Moreover, they referenced how permanent AF induces cerebral hypoperfusion, deposition of $A\beta$ and NFTs and CAA (Tini et al., 2020). These authors included search terms such as 'atrial fibrillation', and 'heart valve disease' which provided them with different search results. Indeed, Benenati et al. (2021) reviewed how AF-induced thromboembolism relates to dementia (Benenati et al., 2021), which was not discussed here in detail. Both of these reviews cite Cacciatore et al. (2012), who linked ventricular rate response in AF to the development of dementia in subjects with mild cognitive impairment (Cacciatore et al., 2012). A recent narrative review of arrhythmias in patients with cardiac amyloidosis weighed the evidence that amyloid protein deposition could contribute to the development of AF (Briasoulis et al., 2023). While amyloid protein deposition is associated with cardiac conduction abnormalities and atrial dilatation (Assaf et al., 2024; Laptseva et al., 2023), and a subset of patients with cardiac amyloidosis who are AF-free at diagnosis go on to develop AF (Sanchis et al., 2019), our review found little direct evidence for a causative role of atrial $A\beta$ protein deposition in the genesis of AF. A recent review by Bazoukis et al. (2024) showed that AF is very prevalent in ATTR, with rates of up to 70% (Bazoukis et al., 2024). Few publications referenced in our review described how ATTR affects cardiac function, and none explored light chain amyloidosis, an important cause of cardiovascular mortality. Similar to our review, Bazoukis et al. (2024) identified how low HRV and autonomic dysregulation increase the risk of AF in cardiac amyloidosis (Bazoukis et al., 2024). Our review has related HRV to amyloidosis in AD. Interestingly, Bazoukis et al. (2024) cited a systematic review investigating AF as a consequence of ATTRwt, by (Mints et al., 2018). They obtained findings of diastolic dysfunction due to cardiac amyloidosis, but the articles they cited related to the incidence and management of AF, not its pathophysiology which was the focus of our review.

**Strengths and limitations.** The major strengths of this review are its systematic nature and the great number of publications it included. Most other reviews focus on one facet of the brain–heart axis, but this review expands upon the cardiogenic dementia hypothesis while

also considering autonomic and hormonal dysregulation. Furthermore, our review found many peer-reviewed publications which strengthens the validity of our search criteria. Indeed, our search was not limited by date of publication, and the use of three separate searches allowed the inclusion of a wide variety of data.

On the other hand, this review has numerous limitations. Firstly, no statistical or sensitivity analysis was performed, which is a key part of the PRISMA framework, meaning that the statistical strength of our findings has not been tested (Page et al., 2021). Secondly, many of the studies included were small and observational, so do not establish causality or determine the direction of association between AD and CVD. Thirdly, despite numerous reviews showing a link between AF and AD, this review did not provide much evidence supporting this. Given that AF is the most common cardiac arrhythmia (Benenati et al., 2021), it is surprising there were few results investigating how AD can predispose to arrhythmias, particularly the effects on the sinoatrial/atrioventricular nodes and ventricular electrophysiology. Similarly, no clinical studies were found to provide evidence of how autonomic dysregulation predisposes to atrial or ventricular arrhythmias. This review highlights existing knowledge gaps in the field. Large-scale, longitudinal cohort studies or randomised controlled trials could help to address these limitations in the future and elucidate the causative role of A$\beta$ amyloidosis in arrhythmogenesis and other cardiac complications of AD.

Even though this systematic review included three separate searches in numerous databases, some of these drawbacks may arise due to our search criteria being narrow. This is supported by key papers being missed by our search. Similarly, Röcken et al. (2002) showed how atrial interstitial amyloidosis and amyloid angiopathy increases the risk of AF (Röcken et al., 2002); this paper did not mention the term 'Alzheimers', so it was not found by our search. Moreover, a meta-analysis missed by our search showed that AD increases the risk of orthostatic hypotension, with an odds ratio of 2.53 (Isik et al., 2022). Inclusion of this study would have provided high-quality evidence for the autonomic dysfunction in AD. Unfortunately, as there are no other systematic reviews on this topic, it is not possible to incorporate search terms from other authors. Future systematic reviews or meta-analyses may overcome this problem by performing bibliography screening of included papers and reviews, to minimise the exclusion of key studies.

**Implications for clinical practice and future directions for research.**   The findings of our review could impact clinical practice. Firstly, the mechanisms detailed here could be applied to develop novel AD therapeutics. For instance, antioxidants could help reduce oxidative stress in cardio-myocytes and endothelial cells or therapeutics may be beneficial through suppressing cortisol or increasing the activity of melatonin. Additionally, non-pharmacological therapies such as breathing techniques to modulate HRV, could reduce cerebral amyloidosis in AD while minimising polypharmacy in patients with comorbidities. Alternatively, this review may introduce new indications for current AD drugs. Aducanumab and Lecanemab, approved for use in AD in 2021 and 2023, respectively, are monoclonal antibodies targeting A$\beta$ plaques (Varadharajan et al., 2023). Given the impact of A$\beta$ amyloidosis on cardiac function, these antibodies may bind to extra-cerebral A$\beta$ and reducing the incidence of cardiovascular complications in AD. Accordingly, multi-centre, randomised, placebo-controlled trials are required to determine the safety and clinical efficacy of these interventions.

Finally, this review showed that CPB during cardiac surgery predisposes to AD pathology and POCD. To increase cardiac and cerebral tolerance to CPB, remote ischaemic preconditioning can be used where transient ischaemia to non-vital tissues can increase protection in other organs. The evidence varies regarding its efficacy: one meta-analysis demonstrated it had no effect on POCD after cardiac surgery (Siburian et al., 2024), whereas a randomised trial concluded a reduction in S100$\beta$ release, indicating neuroprotection (Zhu, Zheng et al., 2022). After conducting our review, additional literature searching yielded no trials relating preconditioning to AD. Therefore, future studies may clarify whether preconditioning could be applied to patients diagnosed with AD to increase their tolerance to subsequent CVDs. However, this is hypothetical and requires proof-of-concept animal studies before human trials are performed.

## Conclusion

In summary, cerebral hypoperfusion increases cognitive dysfunction and predisposes to AD pathology. Cardiac ATTR associated with AD and the development of cardiac dysfunction. Also, pre-amyloid oligomers were found to increase cardiac remodelling seen in AF and heart failure and associated with the development of cardiomyopathies. A$\beta$ and tau amyloidosis were shown to alter cardiomyocyte physiology and impair vasodilatation preclinically and in patients. AD genes also showed this effect, but future work is required to clarify their role in the cardiovascular complications of AD. Impairment of the PNS and SNS potentially predisposes to arrhythmogenesis and impaired blood pressure regulation. Finally, endocrinopathy consisting of hypercortisolism and low melatonin likely contributes to cardiovascular complications of AD.

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

# Additional information

## Competing interests

No competing interests declared.

## Author contributions

S.P.: Conception or design of the work; Drafting the work or revising it critically for important intellectual content; Final approval of the version to be published; Agreement to be accountable for all aspects of the work. A.J.: Conception or design of the work; Drafting the work or revising it critically for important intellectual content; Final approval of the version to be published; Agreement to be accountable for all aspects of the work. S.M.: Conception or design of the work; Drafting the work or revising it critically for important intellectual content; Final approval of the version to be published; Agreement to be accountable for all aspects of the work.

## Funding

British Heart Foundation (BHF): Svetlana Mastitskaya, FS/IBSRF/21/25060.

## Keywords

Alzheimer's disease, amyloidosis, arrhythmia, autonomic dysfunction, cardiovascular disease, hormonal dysfunction

## Supporting information

Additional supporting information can be found online in the Supporting Information section at the end of the HTML view of the article. Supporting information files available:

**Peer Review History**
Supplementary Table – A summary of the publications used in our systematic review.

