## [Peer Review History · The Journal of Physiology]

The causative role of amyloidosis in the cardiac complications of Alzheimer's disease: a comprehensive systematic review

Samuel Parker, Andrew F James, and Svetlana Mastitskaya

DOI: 10.1113/JP286599

Corresponding author(s): Svetlana Mastitskaya (svetlana.mastitskaya@bristol.ac.uk)

Review Timeline:

Submission Date:	01-Dec-2024
Editorial Decision:	27-Jan-2025
Revision Received:	17-Mar-2025
Accepted:	02-Apr-2025

Senior Editor: Bjorn Knollmann

Reviewing Editor: T Alexander Quinn

Transaction Report:

Dear Dr Mastitskaya,

Re: JP-TR-2024-286599 "The causative role of amyloidosis in cardiac complications of Alzheimer's disease: a comprehensive systematic review" by Samuel Parker, Andrew F James, and Svetlana Mastitskaya

Thank you for submitting your manuscript to The Journal of Physiology. It has been assessed by a Reviewing Editor and by 2 expert referees and we are pleased to tell you that it is acceptable for publication following satisfactory revision.

ABSTRACT FIGURES: Authors may use The Journal's premium BioRender account to create/redraw their Abstract Figures (and any other suitable schematic figure). Information on how to access this account is here: <https://physoc.onlinelibrary.wiley.com/journal/14697793/biorender-access>.

REVISION CHECKLIST: Upload a full Response to Referees file. To create your 'Response to Referees' copy all the reports, including any comments from the Senior and Reviewing Editors, into a Microsoft Word, or similar, file and respond to each point, using font or background colour to distinguish comments and responses and upload as the required file type.

We look forward to receiving your revised submission.

Yours sincerely,

Bjorn Knollmann
Senior Editor

EDITOR COMMENTS

Reviewing Editor:

Your paper has been reviewed by two experts in the field, who both felt it is a well-written, comprehensive review, with potential to make an important contribution to the literature. However, they had suggestions to improve the manuscript, which need to be addressed. Please revise the manuscript accordingly, including a point-by-point response to the reviewers' suggestions.

Senior Editor:

I concur with the reviewing editor's assessment.

REFEREE COMMENTS

Referee #1:

The authors have produced a very comprehensive review around amyloidosis, AD and cardiac function. The review is good however there are some key aspects which need to be addressed.

MAJOR

- More and better figures to help the readers understand the processes the authors are trying to convey. There is currently only two figures and the one demonstrating the APP processing pathway(s) is very crude.

- Abeta comes in several species, primarily Abeta 40 and 42. The authors jump between these and sometimes don't refer to which species they are discussing. Also it is often unclear what the authors are trying to demonstrate when they are using the Abeta 40/42 (or 42/40) ratio. This is essential as the different species of Abeta have been shown to have different biological effects. This needs to be addressed and the review needs to be amended accordingly.

- The authors highlight that there are sex specific effects, (eg line 1411, section 4.1.3) however each time these are just comments with no reference as to which sex has the predominant effect.

- APOE ko mice have been shown to be a model of AD. This should be referred to by the authors.

- Authors should make it clear if they are referring to changes in protein/gene or enzymatic activity when they are commenting on APP, BACE1, PSEN1 etc.

- To make the review more impactful this reviewer would suggest the authors to have a section on next steps for researchers to focus on.

MINOR

- The review needs to be checked for grammar. Several sections have words missing in sentences, eg line 537 and section title 4.3.3.

- Genes/proteins need defining, eg line 454 NPPB and 541 NLRP3
- The authors refer to the BACE1 KI mice, however do not demonstrate that the opposite diabetic phenotype is seen in the BACE1 KO mice.
- Section 4.6.2 refers to BDNF . The authors should think about referring to recent studies where this relationship has been shown in other inflammatory/fibrotic diseases (Wasson et al).

Referee #2:

This systematic review makes an important contribution to the growing field of research exploring the connection between amyloidosis in Alzheimer's disease and cardiovascular complications. While it is well-structured, comprehensive, and clinically relevant, addressing the identified pitfalls through broader search criteria, including statistical analyses, and further exploring arrhythmias, molecular mechanisms, and preclinical evidence would improve its scientific rigor and clinical applicability. Below are outlined the major strengths and weaknesses of the study, along with some suggestions to improve the trial.

Strengths:

1. Comprehensive Review:

o The article provides a thorough examination of the relationship between amyloidosis and cardiovascular complications in Alzheimer's disease (AD), covering multiple dimensions such as cerebral and cardiac amyloidosis, autonomic dysfunction, and endocrinopathy. It is an ambitious attempt to collate these aspects, making it an important contribution to understanding the intersection of AD and cardiovascular disease (CVD).

2. Wide Scope of Literature:

o The systematic approach to reviewing studies from multiple databases like PubMed, Ovid Embase/Medline, and CINAHL is a major strength. This increases the review's comprehensiveness and ensures a broader inclusion of relevant studies, strengthening its conclusions.

3. Clear Structure and Logic:

o The introduction sets up the background well, explaining AD and its traditional pathophysiology. The transition to the cardiogenic dementia hypothesis is seamless, providing a logical flow that is easy to follow. The organization of the paper into sections like objectives, discussion, and limitations helps break down complex information for readers.

4. Identification of Gaps and Future Directions:

o The review does a great job of identifying current gaps in the literature, especially in areas like atrial fibrillation (AF) and its role in AD, and the potential role of existing AD drugs in treating cardiovascular complications. This is critical for guiding future research efforts.

5. Relevance to Clinical Practice:

o The review does well in discussing potential clinical implications, particularly the development of targeted therapies, such as antioxidants or modulation of cortisol levels, for mitigating the cardiovascular effects of AD. This practical approach is valuable for researchers and clinicians alike.

Weaknesses and Areas for Improvement:

1. Lack of Statistical Analysis:

o As noted, the absence of a statistical or sensitivity analysis undermines some of the review's rigor. A meta-analysis or quantitative synthesis of the studies would have provided more concrete evidence regarding the strength of the relationships discussed. Incorporating such analyses would align the review with the PRISMA guidelines and bolster its scientific reliability.

2. Limited Discussion on Atrial Fibrillation (AF) and Arrhythmias:

o While the review acknowledges the relationship between AD and AF, the evidence presented is limited. Given that AF is highly prevalent in AD patients, a deeper exploration of the pathophysiology of AF in AD and its interaction with amyloidosis

would have been valuable. Additionally, exploring other arrhythmias related to AD, like ventricular arrhythmias, would have enriched the discussion.

3. Over-reliance on Observational Studies:

o The paper acknowledges that many of the included studies are observational and therefore do not establish causality. This limitation needs more emphasis throughout the discussion, with a clearer indication of how future research should address these methodological concerns (e.g., through large-scale, longitudinal cohort studies or randomized controlled trials).

4. Narrow Search Criteria:

o The review mentions that its search criteria may have been too specific, excluding key papers. Expanding the search terms (e.g., including "heart failure" and "atrial fibrillation") could have potentially uncovered additional relevant studies, especially those examining the interplay between amyloidosis and cardiovascular diseases like AF and atherosclerosis.

5. Confusion in Terminology and Concepts:

o Some terms and concepts, such as the relationship between amyloid beta (A β) and endothelial dysfunction, could benefit from more clarity. For example, the review describes the role of A β in reducing cerebral blood flow through capillary constriction, but this mechanism could be elaborated further with more detail on the molecular pathways involved. Also, the link between A β and other cardiovascular risk factors (like hypertension) could have been discussed in more detail to present a more cohesive narrative.

6. Limited Preclinical Evidence on Arrhythmogenesis:

o The review highlights preclinical evidence regarding the impact of β -secretase on cardiac repolarization, but it does not sufficiently address the conflicting nature of clinical evidence on arrhythmogenesis. More discussion on the discrepancies between preclinical and clinical data would help contextualize the existing literature and give a clearer picture of where further research is needed.

7. Repetition in Some Sections:

o Some points, particularly those related to the bidirectional relationship between AD and CVD, are repeated in both the introduction and the discussion sections. Consolidating these ideas and avoiding redundancy would enhance the paper's readability and conciseness.

Suggestions for Improvement:

1. Incorporate Meta-analysis or Statistical Synthesis:

o A quantitative analysis, such as a meta-analysis, could enhance the scientific rigor of the review and offer more robust conclusions about the relationships between amyloidosis and cardiovascular complications in AD.

2. Expand the Search Terms:

o Consider broadening the search terms to capture studies on a wider range of cardiovascular complications associated with AD (e.g., atrial fibrillation, heart failure, and atherosclerosis). This would also help overcome the limitation of key studies being missed.

3. Clarify the Molecular Mechanisms:

o Provide a more detailed explanation of the molecular pathways linking amyloidosis to cardiovascular dysfunction, especially in terms of oxidative stress, endothelial dysfunction, and the roles of key molecules like endothelin-1 and nitric oxide.

4. Explore More Clinical Evidence on AF and Arrhythmogenesis:

o Given the high prevalence of AF in AD patients, further investigation into its pathophysiology and the role of amyloidosis in arrhythmogenesis is needed. The clinical evidence on the connection between AD and arrhythmias should be expanded to support the review's findings.

5. Address Study Limitations in Greater Depth:

o While the review rightly identifies the observational nature of many studies as a limitation, providing more detail on how these studies could be improved, such as through randomized controlled trials or large cohort studies, would provide clearer

direction for future research.

6. Consider More on Endocrinopathy:

o The role of endocrinopathy in AD-related cardiovascular complications could be discussed further, particularly in relation to the hypothalamic-pituitary-adrenal (HPA) axis dysfunction and the effects of cortisol. More studies linking these hormonal changes to cardiovascular dysfunction could strengthen this section.

7. Streamline the Text:

o Reducing repetitive content and consolidating key points will improve the readability and impact of the review. Consider focusing on the most critical findings to maintain clarity.

REQUIRED ITEMS

- Please include an Abstract Figure file, as well as the Figure Legend text within the main article file. The Abstract Figure is a piece of artwork designed to give readers an immediate understanding of the Review Article and should summarise the main conclusions. If possible, the image should be easily 'readable' from left to right or top to bottom. It should show the physiological relevance of the Review so readers can assess the importance and content of the article. Abstract Figures should not merely recapitulate other figures in the Review. Please try to keep the diagram as simple as possible and without superfluous information that may distract from the main conclusion of the Review. Abstract Figures must be provided by authors no later than the revised manuscript stage and should be uploaded as a separate file during online submission labelled as File Type 'Abstract Figure'. Please ensure that you include the figure legend in the main article file. All Abstract Figures will be sent to a professional illustrator for redrawing and you may be asked to approve the redrawn figure before your paper is accepted.

- Your MS must include a complete "Additional information section" with the following 4 headings and content:

Competing Interests: A statement regarding competing interests. If there are no competing interests, a statement to this effect must be included. All authors should disclose any conflict of interest in accordance with journal policy.

Author contributions: Each author should take responsibility for a particular section of the study and have contributed to writing the paper. Acquisition of funding, administrative support or the collection of data alone does not justify authorship; these contributions to the study should be listed in the Acknowledgements. Additional information such as 'X and Y have contributed equally to this work' may be added as a footnote on the title page.

It must be stated that all authors approved the final version of the manuscript and that all persons designated as authors qualify for authorship, and all those who qualify for authorship are listed.

Funding: Authors must indicate all sources of funding, including grant numbers. If authors have not received funding, this must be stated.

It is the responsibility of authors funded by RCUK to adhere to their policy regarding funding sources and underlying research material. The policy requires funding information to be included within the acknowledgement section of a paper. Guidance on how to acknowledge funding information is provided by the Research Information Network. The policy also requires all research papers, if applicable, to include a statement on how any underlying research materials, such as data, samples or models, can be accessed. However, the policy does not require that the data must be made open. If there are considered to be good or compelling reasons to protect access to the data, for example commercial confidentiality or legitimate sensitivities around data derived from potentially identifiable human participants, these should be included in the statement.

Acknowledgements: Acknowledgements should be the minimum consistent with courtesy. The wording of acknowledgements of scientific assistance or advice must have been seen and approved by the persons concerned. This

section should not include details of funding.

- Please upload separate high quality figure files via the submission form.

- Author profile(s) must be uploaded via the submission form. Authors should submit a short biography (no more than 100 words for one author or 150 words in total for two authors) and a portrait photograph of the two leading authors on the paper. These should be uploaded and clearly labelled together in a Word document with the revised version of the manuscript. Any standard image format for the photograph is acceptable, but the resolution should be at least 300 DPI and preferably more. A group photograph of all authors is also acceptable, providing the biography for the whole group does not exceed 150 words.

- Please include an Abstract Figure file, as well as the Figure Legend text within the main article file. The Abstract Figure is a piece of artwork designed to give readers an immediate understanding of the Review Article and should summarise the main conclusions. If possible, the image should be easily 'readable' from left to right or top to bottom. It should show the physiological relevance of the Review so readers can assess the importance and content of the article. Abstract Figures should not merely recapitulate other figures in the Review. Please try to keep the diagram as simple as possible and without superfluous information that may distract from the main conclusion of the Review. Abstract Figures must be provided by authors no later than the revised manuscript stage and should be uploaded as a separate file during online submission labelled as File Type 'Abstract Figure'. Please ensure that you include the figure legend in the main article file. All Abstract Figures will be sent to a professional illustrator for redrawing and you may be asked to approve the redrawn figure before your paper is accepted.

- Your MS must include a complete "Additional information section" with the following 4 headings and content:

Competing Interests: A statement regarding competing interests. If there are no competing interests, a statement to this effect must be included. All authors should disclose any conflict of interest in accordance with journal policy.

Author contributions: Each author should take responsibility for a particular section of the study and have contributed to writing the paper. Acquisition of funding, administrative support or the collection of data alone does not justify authorship; these contributions to the study should be listed in the Acknowledgements. Additional information such as 'X and Y have contributed equally to this work' may be added as a footnote on the title page.

It must be stated that all authors approved the final version of the manuscript and that all persons designated as authors qualify for authorship, and all those who qualify for authorship are listed.

Funding: Authors must indicate all sources of funding, including grant numbers. If authors have not received funding, this must be stated.

It is the responsibility of authors funded by RCUK to adhere to their policy regarding funding sources and underlying research material. The policy requires funding information to be included within the acknowledgement section of a paper. Guidance on how to acknowledge funding information is provided by the Research Information Network. The policy also requires all research papers, if applicable, to include a statement on how any underlying research materials, such as data, samples or models, can be accessed. However, the policy does not require that the data must be made open. If there are considered to be good or compelling reasons to protect access to the data, for example commercial confidentiality or legitimate sensitivities around data derived from potentially identifiable human participants, these should be included in the statement.

Acknowledgements: Acknowledgements should be the minimum consistent with courtesy. The wording of acknowledgements of scientific assistance or advice must have been seen and approved by the persons concerned. This section should not include details of funding.

- Please upload separate high quality figure files via the submission form.

- Author profile(s) must be uploaded via the submission form. Authors should submit a short biography (no more than 100 words for one author or 150 words in total for two authors) and a portrait photograph of the two leading authors on the paper. These should be uploaded and clearly labelled together in a Word document with the revised version of the manuscript. Any standard image format for the photograph is acceptable, but the resolution should be at least 300 DPI and preferably more. A group photograph of all authors is also acceptable, providing the biography for the whole group does not exceed 150 words.

END OF COMMENTS

MS # JP-TR-2024-286599
Responses to referees' comments

We are grateful for the constructive comments of both reviewers of our original submission to the *J Physiology* and have taken full account of the raised criticisms. We are delighted to have an opportunity to re-submit our work. We now include improved quality figures requested by the reviewers and provide a full response to their comments as well as a thoroughly revised manuscript.

Below we state the criticisms ("critique") and then provide our responses.

Referee #1

The authors have produced a very comprehensive review around amyloidosis, AD and cardiac function. The review is good however there are some key aspects which need to be addressed.

Response: We would like to thank the referee for the time they have taken to review our manuscript and the overall positive assessment of our work. We now include additional figures, provide our responses to all the criticisms raised and submit a thoroughly revised manuscript.

MAJOR critiques

Critique 1: More and better figures to help the readers understand the processes the authors are trying to convey. There is currently only two figures and the one demonstrating the APP processing pathway(s) is very crude.

Response: Thank you for your critique, we agree that the previous figures were simplistic. We have now revised the original figures to meet publication quality standards and added a graphical abstract to enhance the clarity of the manuscript.

Critique 2: Abeta comes in several species, primarily Abeta 40 and 42. The authors jump between these and sometimes don't refer to which species they are discussing. Also it is often unclear what the authors are trying to demonstrate when they are using the Abeta 40/42 (or 42/40) ratio. This is essential as the different species of Abeta have been shown to have different biological effects. This needs to be addressed and the review needs to be amended accordingly.

Response: Thank you for your critique. We agree that we often did not clarify which A β species we were referring to. At some points in our review, we referred to A β generally as studies cited referred to different A β species. Therefore, our revised manuscript identifies the species where specified by the cited paper.

Critique 3: The authors highlight that there are sex specific effects, (eg line 1411, section 4.1.3) however each time these are just comments with no reference as to which sex has the predominant effect.

Response: We agree that at times the affected sex was not specified during our review, and this may have resulted in some confusion regarding our conclusions. We have now clarified the sex-specific findings in more detail.

Critique 4: APOE ko mice have been shown to be a model of AD. This should be referred to by the authors.

Response: Thank you for this valid critique. We did not refer to the APOE KO mouse model of AD originally as no cited studies used this model to draw conclusions. We have now acknowledged that an APOE model would be useful to study the interaction between amyloidosis, AD and cardiovascular complications in section 4.5.

Critique 5: Authors should make it clear if they are referring to changes in protein/gene or enzymatic activity when they are commenting on APP, BACE1, PSEN1 etc.

Response: Thank you for your comment. We have made it clearer when commenting on these species to refer to changes in protein levels or gene expression.

Critique 6: To make the review more impactful this reviewer would suggest the authors to have a section on next steps for researchers to focus on.

Response: Please see section 5.4 where we discussed required areas for future research.

MINOR critiques

Critique: The review needs to be checked for grammar. Several sections have words missing in sentences, eg line 537 and section title 4.3.3.

Response: Thank you for the comment. We have modified text and the section title in question according to the reviewer's comment.

Critique: Genes/proteins need defining, eg line 454 NPPB and 541 NLRP3.

Response: Thank you for the comment. The definitions specifically requested by the reviewer have been provided and the revised manuscript includes definitions of genes/proteins, both in the abbreviations list and at first use in the text.

Critique: The authors refer to the BACE1 KI mice, however do not demonstrate that the opposite diabetic phenotype is seen in the BACE1 KO mice.

Response: Thank you for this comment. We have now acknowledged that BACE1 knockout results in a phenotype that is protective against diabetes and obesity in mice in section 4.3.3. We did not focus on BACE1 KI and KO mice as this was not a topic explored in-depth by many papers cited by our review.

Critique: Section 4.6.2 refers to BDNF. The authors should think about referring to recent studies where this relationship has been shown in other inflammatory/fibrotic diseases (Wasson et al).

Response: Thank you for the comment. We have considered this suggestion but, while the reference suggested is interesting, its inclusion here would represent a significant departure from the focus of our review.

Referee #2:

This systematic review makes an important contribution to the growing field of research exploring the connection between amyloidosis in Alzheimer's disease and cardiovascular complications. While it is well-structured, comprehensive, and clinically relevant, addressing the identified pitfalls through broader search criteria, including statistical analyses, and further exploring arrhythmias, molecular mechanisms, and preclinical evidence would improve its scientific rigor and clinical applicability. Below are outlined the major strengths and weaknesses of the study, along with some suggestions to improve the trial.

Strengths:

1. Comprehensive Review:

The article provides a thorough examination of the relationship between amyloidosis and cardiovascular complications in Alzheimer's disease (AD), covering multiple

dimensions such as cerebral and cardiac amyloidosis, autonomic dysfunction, and endocrinopathy. It is an ambitious attempt to collate these aspects, making it an important contribution to understanding the intersection of AD and cardiovascular disease (CVD).

2. Wide Scope of Literature:

The systematic approach to reviewing studies from multiple databases like PubMed, Ovid Embase/Medline, and CINAHL is a major strength. This increases the review's comprehensiveness and ensures a broader inclusion of relevant studies, strengthening its conclusions.

3. Clear Structure and Logic:

The introduction sets up the background well, explaining AD and its traditional pathophysiology. The transition to the cardiogenic dementia hypothesis is seamless, providing a logical flow that is easy to follow. The organization of the paper into sections like objectives, discussion, and limitations helps break down complex information for readers.

4. Identification of Gaps and Future Directions:

The review does a great job of identifying current gaps in the literature, especially in areas like atrial fibrillation (AF) and its role in AD, and the potential role of existing AD drugs in treating cardiovascular complications. This is critical for guiding future research efforts.

5. Relevance to Clinical Practice:

The review does well in discussing potential clinical implications, particularly the development of targeted therapies, such as antioxidants or modulation of cortisol levels, for mitigating the cardiovascular effects of AD. This practical approach is valuable for researchers and clinicians alike.

Response: We would like to thank this referee for the time taken to review our manuscript and the overall positive assessment of our work. We now provide our responses to all the criticisms raised and submit a thoroughly revised manuscript.

Weaknesses and Areas for Improvement:

Critique 1: Lack of Statistical Analysis.

As noted, the absence of a statistical or sensitivity analysis undermines some of the review's rigor. A meta-analysis or quantitative synthesis of the studies would have provided more concrete evidence regarding the strength of the relationships discussed. Incorporating such analyses would align the review with the PRISMA guidelines and bolster its scientific reliability.

Response: Thank you for your insightful comment. We acknowledge the importance of statistical and sensitivity analyses in enhancing the rigour of systematic reviews. However, our review is not focused on quantitative outcomes or specific measures that would allow for a meta-analysis. Given the broad and multifaceted nature of the brain-heart axis in Alzheimer's disease, our approach was to synthesise existing knowledge across various interconnected domains rather than focus on a single measurable outcome.

While a meta-analysis would have been feasible if we had limited our scope – for instance, by examining plasma A β levels as a predictive marker for cardiac complications – this would have significantly narrowed the breadth of our review, reducing the included studies from 252 to approximately 20-30. Instead, our goal was

to provide a comprehensive overview of the complex interactions between the brain and heart in Alzheimer's disease, which we consider a key strength of our work. We appreciate the suggestion and have clarified this point in the revised manuscript to emphasize the rationale behind our methodological choices.

Critique 2: Limited Discussion on Atrial Fibrillation (AF) and Arrhythmias.

While the review acknowledges the relationship between AD and AF, the evidence presented is limited. Given that AF is highly prevalent in AD patients, a deeper exploration of the pathophysiology of AF in AD and its interaction with amyloidosis would have been valuable. Additionally, exploring other arrhythmias related to AD, like ventricular arrhythmias, would have enriched the discussion.

Response: Given the association of cardiac amyloidosis with conduction disorders and atrial remodelling, and the association of fibrosis generally with a substrate for atrial fibrillation, we were indeed expecting to uncover evidence for a causative role of A β -amyloidosis in arrhythmogenesis. To our disappointment, we did not find sufficient evidence of arrhythmogenesis from our searches. In this systematic review, we have reported the results of our findings with the search terms used. While the causative role of cardiac arrhythmias in cerebral hypoperfusion and, consequently, accelerated AD progression is unquestionable, the evidence for a reverse association (increased levels of A β causing cardiac arrhythmias) is limited. We have clarified this point in Section 5.1, have added new text, with references, to Section 5.2 of the Discussion, and stress the need for exploring pathophysiology of AF and other arrhythmias in AD in Section 5.3.

Critique 3: Over-reliance on Observational Studies.

The paper acknowledges that many of the included studies are observational and therefore do not establish causality. This limitation needs more emphasis throughout the discussion, with a clearer indication of how future research should address these methodological concerns (e.g., through large-scale, longitudinal cohort studies or randomized controlled trials).

Response: The purpose of this systematic review was to compile the existing evidence and identify knowledge gaps. Our search terms yielded mostly observational data – this reflects the current state of the field and highlights gaps in knowledge. As suggested, we have included a sentence emphasizing the necessity for large-scale, longitudinal cohort studies and randomized controlled trials in Section 5.3 of the Discussion (paragraph 2).

Critique 4: Narrow Search Criteria.

The review mentions that its search criteria may have been too specific, excluding key papers. Expanding the search terms (e.g., including "heart failure" and "atrial fibrillation") could have potentially uncovered additional relevant studies, especially those examining the interplay between amyloidosis and cardiovascular diseases like AF and atherosclerosis.

Response: We acknowledge this limitation in our text (Section 5.2). We did our best to cast the net wide and capture all cardiac-related complications associated with beta-amyloidosis, by including search terms "cardiac*" or "heart" (which would cover "heart failure") and "arrhythm*" (which was anticipated to cover all kinds of arrhythmias, including atrial fibrillation). After writing the first draft, we realised that some papers were not captured, so we added more search terms and repeated the screening from the beginning (which delayed us by another three months). Unfortunately, there is always a risk of not capturing some publications, even if it's indefinitely low.

Critique 5: Confusion in Terminology and Concepts.

Some terms and concepts, such as the relationship between amyloid beta (A β) and endothelial dysfunction, could benefit from more clarity. For example, the review describes the role of A β in reducing cerebral blood flow through capillary constriction, but this mechanism could be elaborated further with more detail on the molecular pathways involved. Also, the link between A β and other cardiovascular risk factors (like hypertension) could have been discussed in more detail to present a more cohesive narrative.

Response: Thank you for this comment. Initially, we tried to base the discussion primarily on the papers identified through our search. However, while hypertension may exacerbate A β amyloidosis and cognitive impairment (discussed in detail in section 4.1.6), the evidence of a reverse causal effect – namely, that A β contributes to the development of hypertension, is quite limited. Our search did identify a few studies showing that A β increases ET1 levels through ROS production, toxicity to endothelial cells and impaired NOS, and we discussed them in section 4.3.2, as well as in the first paragraph of the discussion. In the revised version, we expand on the ET1-mediated mechanisms of vasoconstriction (Discussion, 1st paragraph). We have also added a sentence on the potential causative link between A β and hypertension through increased ET1 levels, highlighting that this possibility warrants further investigation.

Critique 6: Limited Preclinical Evidence on Arrhythmogenesis.

The review highlights preclinical evidence regarding the impact of β -secretase on cardiac repolarization, but it does not sufficiently address the conflicting nature of clinical evidence on arrhythmogenesis. More discussion on the discrepancies between preclinical and clinical data would help contextualize the existing literature and give a clearer picture of where further research is needed.

Response: As per our response to Critique 2, we were really interested to find the evidence for arrhythmogenesis. There is not sufficient preclinical evidence, which highlights the knowledge gap.

Critique 7: Repetition in Some Sections.

Some points, particularly those related to the bidirectional relationship between AD and CVD, are repeated in both the introduction and the discussion sections. Consolidating these ideas and avoiding redundancy would enhance the paper's readability and conciseness.

Response: Thank you for this comment. We now have revised the text of introduction and Discussion accordingly.

Suggestions for Improvement:

Suggestion 1: Incorporate Meta-analysis or Statistical Synthesis.

A quantitative analysis, such as a meta-analysis, could enhance the scientific rigor of the review and offer more robust conclusions about the relationships between amyloidosis and cardiovascular complications in AD.

Response: Thank you for this suggestion. Unfortunately, this is not feasible, given the qualitative nature of the findings reported in the papers identified by our search.

Suggestion 2. Expand the Search Terms:

Consider broadening the search terms to capture studies on a wider range of cardiovascular complications associated with AD (e.g., atrial fibrillation, heart failure, and atherosclerosis). This would also help overcome the limitation of key studies being missed.

Response: As per our response to Critique 2, the search terms suggested by the reviewer are already covered by our broader search terms (such as “heart”, “arrhythm*”). Repeating the search with a new set of search terms isn’t feasible as it would cause an unnecessary delay (we have already performed two rounds of search, each screening taking 3-4 months).

Suggestion 3. Clarify the Molecular Mechanisms:

Provide a more detailed explanation of the molecular pathways linking amyloidosis to cardiovascular dysfunction, especially in terms of oxidative stress, endothelial dysfunction, and the roles of key molecules like endothelin-1 and nitric oxide.

Response: The purpose of this systematic review was to identify the existing evidence for the causative role of A β amyloidosis in the cardiovascular complications of AD. We believe we have explained the mechanisms in sufficient detail within the scope of the review, with references to original studies that will direct interested readers to the relevant sources. Additionally, we have expanded on the endothelial dysfunction-ET1-ROS-vasoconstriction link in the revised version of the manuscript, following the reviewer’s suggestion.

Suggestion 4. Explore More Clinical Evidence on AF and Arrhythmogenesis:

Given the high prevalence of AF in AD patients, further investigation into its pathophysiology and the role of amyloidosis in arrhythmogenesis is needed. The clinical evidence on the connection between AD and arrhythmias should be expanded to support the review's findings.

Response: We wish there was more preclinical and clinical evidence on causative role of A β in arrhythmogenesis. As per our response to critique 2, we have reported the results of our searches, and evidence for a causal relation between AD and AF is very limited.

Suggestion 5. Address Study Limitations in Greater Depth:

While the review rightly identifies the observational nature of many studies as a limitation, providing more detail on how these studies could be improved, such as through randomized controlled trials or large cohort studies, would provide clearer direction for future research.

Response: Thank you, we now have highlighted the need for detailed quantitative research in this area and included suggestions on how these studies can be improved.

Suggestion 6. Consider More on Endocrinopathy:

The role of endocrinopathy in AD-related cardiovascular complications could be discussed further, particularly in relation to the hypothalamic-pituitary-adrenal (HPA) axis dysfunction and the effects of cortisol. More studies linking these hormonal changes to cardiovascular dysfunction could strengthen this section.

Response: Thank you for the comment. While in the original version of the manuscript there is a full section (4.7.1 HPA axis dysfunction relates to cardiovascular sequelae of Alzheimer’s disease), amounting to two pages, that covers this topic, we agree that extra text in the Discussion section (Section 5.1) would be helpful, and have therefore expanded this section.

Suggestion 7. Streamline the Text:

Reducing repetitive content and consolidating key points will improve the readability and impact of the review. Consider focusing on the most critical findings to maintain clarity.

Response: Thank you for this suggestion. We now have revised the text of Introduction and Discussion accordingly.

Dear Dr Mastitskaya,

Re: JP-TR-2025-286599R1 "The causative role of amyloidosis in the cardiac complications of Alzheimer's disease: a comprehensive systematic review" by Samuel Parker, Andrew F James, and Svetlana Mastitskaya

We are pleased to tell you that your paper has been accepted for publication in The Journal of Physiology.

Authors should note that it is too late at this point to offer corrections prior to proofing. Major corrections at proof stage, such as changes to figures, will be referred to the Editors for approval before they can be incorporated. Only minor changes, such as to style and consistency, should be made at proof stage. Changes that need to be made after proof stage will usually require a formal correction notice.

Yours sincerely,

Bjorn Knollmann
Senior Editor
The Journal of Physiology

P.S. - You can help your research get the attention it deserves! Check out Wiley's free Promotion Guide for best-practice recommendations for promoting your work at www.wileyauthors.com/eoo/guide. You can learn more about Wiley Editing Services which offers professional video, design, and writing services to create shareable video abstracts, infographics, conference posters, lay summaries, and research news stories for your research at www.wileyauthors.com/eoo/promotion.

IMPORTANT NOTICE ABOUT OPEN ACCESS: To assist authors whose funding agencies mandate public access to published research findings sooner than 12 months after publication, The Journal of Physiology allows authors to pay an Open Access (OA) fee to have their papers made freely available immediately on publication.

You can check if your funder or institution has a Wiley Open Access Account here: <https://authorservices.wiley.com/author-resources/Journal-Authors/licensing-and-open-access/open-access/author-compliance-tool.html>.

EDITOR COMMENTS

Reviewing Editor:

The authors have adequately addressed the reviewers' concerns.

Senior Editor:

The MS is now acceptable for publication. Thank you for contributing this interesting article!

REFEREE COMMENTS

Referee #1:

The authors have adequately addressed all the comments that were made. I am happy for this review to be published.

Referee #2:

After a long and careful review of the project, the authors suitably fulfilled the comments of the reviewer, integrating his suggestions in the manuscript. This, the study reached out an high level of impact and influence that allows me to recommend for publication.